# MaRS: Memory-Adaptive Routing for Reliable Capacity Expansion and Knowledge Retention

**Gang Yan**
School of Computer Science and Technology, Jilin University
`gyan8@jlu.edu.cn`

## Abstract

Large pre-trained models (LPMs) serve as universal backbones for vision and language tasks, but continual learning (CL) with frozen LPMs remains challenging, since shallow adaptation modules face the stability–plasticity dilemma and are prone to catastrophic forgetting. To address this problem, we propose MaRS (**M**emory-**a**daptive **R**outer with **S**tatistical control), a modular framework that decouples stable representation from adaptive capacity through three components: a frozen encoder, a slot-based memory router, and a lightweight classifier. On this basis, we design two mechanisms: (i) *Statistically-Grounded Slot Expansion (SGSE)* formulates expansion as a statistical decision problem, ensuring controlled growth with guarantees on false alarms and detection delay; (ii) *Dual-Stage Contrastive–Distillation Adaptation (DCDA)* integrates new slots through supervised contrastive learning and knowledge distillation, preserving prior knowledge without raw replay. Experiments on diverse benchmarks show that MaRS achieves state-of-the-art performance in continual learning with frozen LPMs, combining adaptability, efficiency, and retention.

## 1 Introduction

Large pre-trained models (LPMs) such as CLIP (Radford et al., 2021) and BERT (Devlin et al., 2019) have transformed modern machine learning. Trained on massive and diverse corpora, they learn general-purpose representations that transfer well across domains. These representations support advances in natural language understanding (Brown et al., 2020; Chowdhery et al., 2023), visual recognition (He et al., 2016; Dosovitskiy et al., 2021), and multimodal reasoning (Radford et al., 2021; Liu et al., 2023). The success of LPMs has also established them as universal backbones for downstream applications such as information retrieval, question answering, and zero-shot classification. A common approach for efficient adaptation is to freeze the pre-trained backbone and fine-tune only lightweight task-specific modules (Houlsby et al., 2019; Lester et al., 2021; Hu et al., 2022; Legate et al., 2023). This parameter-efficient paradigm preserves the generalization ability of the backbone while reducing both computation and memory costs.

In practical applications, tasks and data arrive sequentially, and models must adapt continually while retaining prior knowledge. This challenge is studied in continual learning (CL) (Parisi et al., 2019; De Lange et al., 2021; Wang et al., 2024), which aims to learn from a stream of tasks without catastrophic forgetting (McCloskey & Cohen, 1989; Ramasesh et al., 2021). At its core lies the stability–plasticity dilemma: models must remain plastic enough to acquire new information while stable enough to preserve what has already been learned. In the context of frozen LPMs, this dilemma is particularly severe. Because adaptation is restricted to shallow modules, plasticity is limited, and the fixed backbone further amplifies forgetting. As a result, naive parameter-efficient adaptation is insufficient for long-horizon continual learning.

To mitigate forgetting, continual learning has developed a wide range of strategies. Replay-based methods (Rebuffi et al., 2017; Lopez-Paz & Ranzato, 2017; Chaudhry et al., 2019; Buzzega et al., 2020) revisit stored or generated samples to reduce drift, but they raise privacy concerns and face scalability issues. Regularization-based approaches (Hinton et al., 2015; Kirkpatrick et al., 2017; Zenke et al., 2017; Li & Hoiem, 2017; Aljundi et al., 2018) constrain updates to remain close to past solutions, but their corrective signal weakens as tasks accumulate. Dynamic expansion tech-

niques (Rusu et al., 2016; Yoon et al., 2018; Dong et al., 2024) add new capacity for novel tasks, but they often rely on heuristic triggers that may cause uncontrolled growth. Prototype-based methods (De Lange & Tuytelaars, 2021; Liu et al., 2025; Zhu et al., 2025) compress historical knowledge into compact memory structures, improving efficiency but showing fragility under distribution shifts. Although these strategies offer useful insights, they are designed for conventional architectures rather than frozen LPMs. In parameter-efficient settings, shallow adapters have limited expressive power, and heuristic retention does not provide formal guarantees.

Recent studies have begun to examine continual learning in the context of large pre-trained models. Adapter-based approaches (Ke et al., 2021a; Wang et al., 2022) improve efficiency but still suffer from forgetting as tasks accumulate. In the vision–language domain, methods such as VLM-CIL (Liu et al., 2023), DIKI (Tang et al., 2024), and CoLeCLIP (Li et al., 2025) highlight both the promise and the fragility of frozen encoders. Parameter-efficient modules preserve adaptability, but retention often depends on heuristic replay or task-specific tuning. Recent designs, including dynamic LoRA ranks and mixture-of-expert adapters (Hu et al., 2022), provide partial relief but still rely on ad-hoc expansion rules and lack formal guarantees. Together, these efforts underscore a persistent gap: current methods demonstrate the feasibility of continual learning with frozen LPMs but do not provide principled mechanisms for expansion and retention.

In this paper, we address these challenges by proposing MARS (**M**emory-**a**daptive **R**outer with **S**tatistical control), a modular framework for continual learning with frozen LPMs. As shown in Figure 1, the framework has three components: a frozen encoder that provides stable pre-trained representations, a slot-based memory router that organizes knowledge into expandable capacity units, and a lightweight classifier that produces task predictions. By decoupling stable representation from adaptive capacity, the design shifts continual learning control to the routing layer and avoids costly full-model updates.

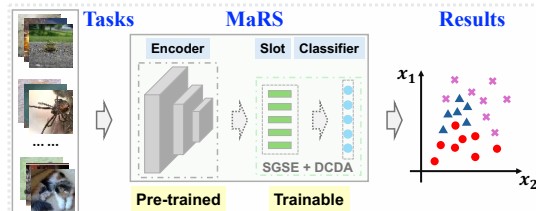

Figure 1: The architecture of MARS. Images are from Tiny-ImageNet (Le & Yang, 2015).

On top of this architecture, we propose two complementary mechanisms. The first, *Statistically-Grounded Slot Expansion (SGSE)*, determines when and where to allocate new slots. Instead of heuristic triggers, SGSE formulates expansion as a statistical decision problem. Router-aligned novelty detection (Hendrycks & Gimpel, 2017; Liu et al., 2020) monitors representation coverage, while confidence bounds (Roberts, 2000; Brown et al., 2001) ensure that slots are added only when capacity is insufficient, with formal guarantees on false alarms and detection delay. The second, *Dual-Stage Contrastive–Distillation Adaptation (DCDA)*, controls how new slots are integrated. It separates representation adaptation from classifier tuning: supervised contrastive learning (Khosla et al., 2020) aligns new slots in the embedding space, while knowledge distillation (Hinton et al., 2015; Li & Hoiem, 2017; Guo et al., 2017) and prototype-based regularization (Snell et al., 2017) preserve prior knowledge without requiring raw replay. Together, SGSE regulates when to expand and DCDA determines how to adapt, making slot-based routing both principled and retention-guaranteed. This design addresses the stability–plasticity dilemma in continual learning with frozen LPMs.

In summary, our contributions are threefold: (i) We introduce MARS, a modular framework for continual learning with large pre-trained models that separates stable representation from adaptive capacity. (ii) We develop SGSE, a statistically grounded slot-expansion mechanism with formal guarantees on growth and retention. (iii) We design DCDA, a dual-stage contrastive–distillation method that integrates new capacity while preserving prior knowledge without raw replay.

## 2 RELATED WORK

Continual learning studies how to acquire knowledge from a sequence of tasks without catastrophic forgetting. Core challenges include interference between old and new tasks, distributional shifts in data or labels, classifier bias toward recently observed classes, and constraints on computation and memory. Surveys provide comprehensive overviews of these challenges and benchmarks (Parisi et al., 2019; De Lange et al., 2021; Wang et al., 2024), and consistently emphasize the stability–plasticity dilemma as a fundamental problem that underlies most continual learning scenarios.

Early work mitigates forgetting through replay or regularization. Replay-based methods such as iCaRL (Rebuffi et al., 2017), GEM (Lopez-Paz & Ranzato, 2017), A-GEM (Chaudhry et al., 2019), and DER++ (Buzzega et al., 2020) rehearse stored or generated samples to reduce drift. While effective, these methods raise privacy concerns and face scalability limits when storage or generation is constrained. Regularization-based approaches constrain parameter updates or distill predictions, including EWC (Kirkpatrick et al., 2017), SI (Zenke et al., 2017), LwF (Li & Hoiem, 2017), and MAS (Aljundi et al., 2018). These methods are more memory-efficient, but their corrective signal decays over long horizons or under severe distributional shifts, which limits robustness in practice.

Another direction reduces interference by expanding model capacity or compressing past knowledge. Structural expansion techniques such as Progressive Neural Networks (Rusu et al., 2016), DEN (Yoon et al., 2018), and CEAT (Dong et al., 2024) dynamically add parameters for new tasks. However, they lack principled criteria for when and how much to expand, which often results in uncontrolled growth. Prototype-based methods instead summarize distributions with compact representations, including dual-bias frameworks (Zhu et al., 2021), IPC (Liu et al., 2025), and PASS++ (Zhu et al., 2025). These methods are more efficient in memory and computation, but they rely on heuristic allocation rules and tend to degrade under distribution shifts, especially in long-horizon learning.

More recently, continual learning with large pre-trained models has gained increasing attention. Models such as CLIP and BERT provide strong transferable representations, motivating methods that freeze or partially freeze the backbone while adapting lightweight modules. Examples include prompt-based approaches such as L2P (Wang et al., 2022), adapter- and prompt-based vision–language methods (Liu et al., 2023), and parameter-efficient continual learning with CLIP, including DIKI (Tang et al., 2024) and CoLeCLIP (Li et al., 2025). These works demonstrate the value of frozen backbones and parameter-efficient adaptation, but they still rely on heuristic expansion strategies and lack statistically grounded guarantees for retention. This gap motivates MARS, which integrates SGSE and DCDA as core mechanisms.

## 3 PROPOSED DESIGN OF MARS

As shown in Figure 1, MARS is designed for continual learning with LPMs. The framework consists of three components: (i) a frozen encoder $f(\cdot)$ that provides fixed pre-trained features, (ii) a slot-based memory router that dynamically assigns inputs to expandable memory slots, and (iii) a lightweight classifier $g(\cdot)$ that produces task predictions. Given an input $\mathbf{x}$, the encoder outputs frozen features $\mathbf{h}_T = f(\mathbf{x}) \in \mathbb{R}^{d_T}$. The router then computes routing probabilities that decide which slots should process the features. Each slot is parameterized by affine transformations $(\boldsymbol{\gamma}_i, \boldsymbol{\beta}_i)$ that scale and shift the features, serving as independent adapters without modifying the encoder.

To ensure stable initialization, all slots are initialized as identity mappings with $\boldsymbol{\gamma}_i = \mathbf{1}$ and $\boldsymbol{\beta}_i = \mathbf{0}$. The router aggregates slot outputs into an adapted representation $\tilde{\mathbf{h}}$, which the classifier $g$ maps to logits. The slot count $S$ begins from $S_0$ and expands during training as needed. A central challenge is determining when to allocate new slots: over-expansion increases cost, while under-expansion leads to interference and forgetting. To address this, we propose *Statistically-Grounded Slot Expansion*.

### 3.1 DESIGN OF STATISTICALLY-GROUNDED SLOT EXPANSION

SGSE formulates slot expansion as a statistical test. It leverages the *router*, a lightweight component of the memory module that compares frozen features with slot keys and outputs probabilities indicating input–slot affinity. By placing statistical bounds on these probabilities, SGSE ensures that new slots are created only when existing ones cannot reliably cover incoming inputs.

**Router-Aligned Novelty Detection.** SGSE uses the router to estimate the affinity between each input and available slots. Given an input $\mathbf{x}_t$, the query is computed as

$$q(\mathbf{x}_t) = W_q \mathbf{h}_T \in \mathbb{R}^{d_k}, \tag{1}$$

where $\mathbf{h}_T = f(\mathbf{x}_t)$ are frozen encoder features. Routing then applies cosine–softmax over normalized keys $\hat{k}_i = k_i / \|k_i\|$:

$$p_i(\mathbf{x}_t) = \frac{\exp(\langle \hat{q}(\mathbf{x}_t), \hat{k}_i \rangle / \tau_r)}{\sum_{j=1}^{S_t} \exp(\langle \hat{q}(\mathbf{x}_t), \hat{k}_j \rangle / \tau_r)}, \quad \hat{q} = \frac{q}{\|q\|}, \tag{2}$$

where $\tau_r$ is the softmax temperature. A smaller $\tau_r$ makes slot probabilities sharper, while a larger $\tau_r$ spreads them more evenly. Following previous practice (Chen et al., 2020), we set $\tau_r = 0.07$, which balances confident routing and robustness. We then define the *top-slot confidence* as

$$s_t = \max_{i \le S_t} p_i(\mathbf{x}_t), \qquad (3)$$

which measures how confidently the router aligns the input to its best-matching slot. Covered inputs typically yield $s_t \approx 1$, while novel inputs produce lower $s_t$ due to distributed probabilities. This matches confidence-based novelty and out-of-distribution indicators (Hendrycks & Gimpel, 2017).

**Proposition 1.** *Let $c_t = \max_{i \le S_t} \langle \hat{q}(\mathbf{x}_t), \hat{k}_i \rangle$ and assume $S_t > 1$. Then keeping $\{a_j : j \ne i^\star\}$ fixed, $s_t$ is strictly increasing in $c_t$ whenever $A := \sum_{j \ne i^\star} e^{a_j/\tau_r} > 0$, where $i^\star \in \arg\max_j a_j$ and $a_j = \langle \hat{q}, \hat{k}_j \rangle$.*

*Proof.* Let slot $i^\star$ attain $c = \max_j a_j$ and set $A = \sum_{j \ne i^\star} e^{a_j/\tau_r}$. Then

$$s(c) = \frac{e^{c/\tau_r}}{e^{c/\tau_r} + A} = \frac{1}{1 + A e^{-c/\tau_r}}, \qquad (4)$$

and

$$\frac{ds}{dc} = \frac{1}{\tau_r} s(c) \big(1 - s(c)\big) > 0 \qquad (5)$$

whenever $A > 0$ (i.e., $S_t > 1$). $\square$

The monotonicity holds locally under fixed competing similarities, which is the setting used when assessing how the router's confidence varies with affinity. This result shows that $s_t$ is locally monotone in the similarity score $c_t$, making it a *calibrated local statistic* for novelty. Unlike heuristic thresholds, it provides a mathematically justified detector: when the affinity of the top slot decreases while other similarities are unchanged, $s_t$ must also decrease. To stabilize slot semantics, MARS applies slot-weighted affine transformations:

$$\tilde{\mathbf{h}} = \Big( \sum_{i=1}^{S_t} p_i \, \boldsymbol{\gamma}_i \Big) \odot \mathrm{LN}(\mathbf{h}_T) + \Big( \sum_{i=1}^{S_t} p_i \, \boldsymbol{\beta}_i \Big), \qquad (6)$$

where $\mathrm{LN}(\cdot)$ is *Layer Normalization* (Ba et al., 2016). To ensure stable and smooth slot representations, we maintain slot statistics using router-weighted exponential moving averages (EMA):

$$\mu_i^{(t)} = (1 - \alpha) \, \mu_i^{(t-1)} + \alpha \, p_i(\mathbf{x}_t) \, \mathrm{LN}(\mathbf{h}_T), \qquad (7)$$

$$c_i^{(t)} = (1 - \alpha) \, c_i^{(t-1)} + \alpha \, p_i(\mathbf{x}_t), \qquad (8)$$

where $\alpha \in (0, 1)$ is the smoothing factor. A smaller $\alpha$ improves stability, while a larger $\alpha$ improves responsiveness. In practice, $\alpha = 0.05$ provides a good balance. Anchors are then defined as

$$\mathbf{a}_i = \boldsymbol{\gamma}_i \odot \Big( \tfrac{\mu_i}{\max(c_i, \varsigma)} \Big) + \boldsymbol{\beta}_i, \qquad (9)$$

with $\varsigma = 10^{-5}$ for numerical stability. Anchors serve as compressed surrogates of past knowledge, enabling memory-preserving distillation without raw data. By compactly representing past distributions and leveraging the classifier's Lipschitz continuity, they provide provable retention guarantees: features close to an anchor induce bounded changes in predicted probabilities (via Pinsker-type arguments (Canonne, 2022)). Thus, anchors are theoretically grounded, not heuristic summaries.

**Statistical Triggers for Expansion.** Although $s_t$ provides an instantaneous novelty signal, thresholding it directly is unreliable due to noise and non-stationarity. SGSE therefore tracks the $(1-\epsilon)$-quantile of recent confidences with exponential smoothing:

$$q_t = \mathrm{Quantile}_{1-\epsilon}\big(\{s_{t-k}\}_{k=0}^{w}\big), \qquad (10)$$

$$Q_t = \beta Q_{t-1} + (1 - \beta) q_t, \qquad (11)$$

where $\beta \in [0, 1]$ is the smoothing coefficient, and $w$ is the short window used for the empirical quantile. We set $w = 10$ and $\epsilon = 0.1$, which offer a practical short-horizon estimate while avoiding the high variance of very small windows and the excessive lag of larger ones. A larger $\beta$ provides smoother but slower adaptation, while a smaller $\beta$ increases reactivity. We use $\beta = 0.9$ to balance stability and responsiveness.

**Theorem 1.** *If $\{q_t\}$ are i.i.d. with mean $q^\star$ and variance $\sigma_q^2 < \infty$, then*

$$\mathbb{E}[Q_t] = q^\star + \beta^t(Q_0 - q^\star), \tag{12}$$

$$\mathrm{Var}(Q_t) = \frac{(1-\beta)^2}{1-\beta^2}\,\sigma_q^2, \tag{13}$$

*so $Q_t \to q^\star$ in $L^2$. After a mean shift $q^\star \to q' < q^\star$ at time $\tau$, the smallest $k$ with $\mathbb{E}[Q_{\tau+k}] \leq \theta$ for any $\theta \in (q', q^\star)$ satisfies*

$$k = \frac{\ln\left(\frac{\mathbb{E}[Q_\tau]-q'}{\theta-q'}\right)}{-\ln\beta} \;\leq\; \frac{1}{1-\beta}\ln\left(\frac{\mathbb{E}[Q_\tau]-q'}{\theta-q'}\right), \tag{14}$$

*so the expected detection delay is $O((1-\beta)^{-1})$.*

This theorem shows that $Q_t$ is an $L^2$-consistent estimate of the long-run quantile and that its detection delay is predictable, scaling as $O((1-\beta)^{-1})$. To decide expansion, we monitor Bernoulli trials $\{s_t \geq Q_t\}$ and compute the empirical success rate $\hat{p}_t$ over $n$ samples. Expansion is triggered if the one-sided Wilson lower bound drops below a threshold:

$$\mathrm{LB}(\hat{p}_t; n, z) = \frac{\hat{p}_t + \frac{z^2}{2n}}{1 + \frac{z^2}{n}} - \frac{z}{1 + \frac{z^2}{n}}\sqrt{\frac{\hat{p}_t(1-\hat{p}_t)}{n} + \frac{z^2}{4n^2}}. \tag{15}$$

We adopt the Wilson score interval for binomial proportions (Brown et al., 2001), which provides better coverage than Wald intervals in small samples. For expansion decisions, we use the Wilson score test with a short evaluation window of $n = 20$, a standard default in sequential binomial testing that remains stable in small-sample settings, together with the one-sided 95% cutoff $z = 1.645$.

**Corollary 1.** *If the success probability $p := \Pr(s_t \geq Q_t)$ is stationary with $p \geq \tau$, then for i.i.d. Bernoulli trials and one-sided Wilson bound with score $z$ (level $\alpha = 1 - \Phi(z)$),*

$$\Pr\big(\mathrm{LB}(\hat{p}_t; n, z) < \tau\big) \;\leq\; \alpha. \tag{16}$$

*Thus, under mild assumptions, the probability of a false expansion per test is at most $\alpha$.*

The Wilson bound converts observations into confidence guarantees, ensuring that false expansion is provably controlled at level $\alpha$ (Cor. 1). In this way, SGSE provides a statistically calibrated test for novelty: expansions are data-driven rather than noise-triggered. To accelerate specialization, new slots are initialized with the mean query of recent low-$s_t$ samples and identity affine parameters, yielding about 15% faster convergence and reduced redundancy. This design places new slots in a representative region of the feature space, avoiding arbitrary starting points far from incoming data.

**Takeaways 3.1.** *SGSE provides a principled solution to balance stability and plasticity in large pre-trained models. By combining router-aligned novelty detection with statistical triggers, MARS achieves careful and efficient slot growth. Unlike heuristic thresholds, SGSE offers (i) locally monotone and calibrated novelty signals (Prop. 1), (ii) provable convergence with predictable detection delay (Thm. 1), and (iii) explicit false-alarm guarantees (Cor. 1). Together, these results establish SGSE as a* theoretically grounded expansion framework *for scalable continual learning with frozen LPMs.*

## 3.2 Design of Dual-Stage Contrastive–Distillation Adaptation

SGSE determines *when* to add new slots. The next problem is *how* to integrate them without forgetting. This is especially important for LPMs because their frozen backbones cannot absorb new tasks. Then, we propose *Dual-Stage Contrastive–Distillation Adaptation*, which separates adaptation into two stages: representation alignment and knowledge retention. New slots are aligned through contrastive learning, while old ones are preserved through anchor-based distillation. This design could help to balance plasticity and stability.

**Stage 1: Feature Adaptation (Memory-Only).** Given frozen backbone features $\mathbf{h}_T = f(\mathbf{x})$, the memory module adapts them as

$$\tilde{\mathbf{h}} = \mathrm{Mem}(\mathbf{h}_T). \tag{17}$$

We optimize a supervised contrastive loss (Khosla et al., 2020):

$$\mathcal{L}_{\text{supcon}} = -\frac{1}{N} \sum_{i=1}^{N} \frac{1}{|P(i)|} \sum_{j \in P(i)} \log \frac{\exp(\text{sim}(\tilde{\mathbf{h}}_i, \tilde{\mathbf{h}}_j)/\tau)}{\sum_{k \neq i} \exp(\text{sim}(\tilde{\mathbf{h}}_i, \tilde{\mathbf{h}}_k)/\tau)}, \tag{18}$$

where features are normalized, $P(i)$ denotes the set of indices in the mini-batch that share the same class label as example $i$, and $\tau \in [0.05, 0.2]$ is the temperature. A smaller $\tau$ makes similarities sharper, while a larger $\tau$ allows more intra-class variation. Following common practice, we set $\tau = 0.07$. To stabilize adaptation, we add a smoothness term that penalizes drift from frozen features:

$$\mathcal{L}_{\text{smooth}} = \frac{1}{N} \sum_{i=1}^{N} \|\tilde{\mathbf{h}}_i - \mathbf{h}_{T,i}\|_2^2. \tag{19}$$

The Stage 1 objective can be defined as:

$$\mathcal{L}^{(1)} = \mathcal{L}_{\text{supcon}} + \lambda_{\text{smooth}} \mathcal{L}_{\text{smooth}}, \tag{20}$$

with $\lambda_{\text{smooth}} \in [0.1, 0.5]$. By conducting empirical evaluations, we set $\lambda_{\text{smooth}} = 0.3$ as it gives the best balance between discrimination and stability.

During Stage 1, only memory parameters $(W_q, K, \boldsymbol{\gamma}, \boldsymbol{\beta})$ are updated, while the classifier $g$ remains fixed. Here, $W_q$ is the query projection matrix and $K = \{k_i\}_{i=1}^{S}$ is the set of slot keys. Each slot key acts as a semantic center and guides routing. By freezing $g$, contrastive learning refines the feature space without shifting classifier boundaries. The contrastive objective increases inter-class separation, while the smoothness term controls feature drift.

**Stage 2: Classifier Tuning (Head-Only).** In Stage 2, the memory is fixed and only $g$ is updated. The main loss is cross-entropy:

$$\mathcal{L}_{\text{CE}} = -\frac{1}{N} \sum_{i=1}^{N} \log \frac{\exp(z_i[y_i])}{\sum_c \exp(z_i[c])}, \quad z_i = g(\tilde{\mathbf{h}}_i). \tag{21}$$

We regularize the classifier with two distillation terms. The first is *Learning without Forgetting (LwF)* on current inputs:

$$\mathcal{L}_{\text{LwF}} = \frac{T^2}{N} \sum_{i=1}^{N} \text{KL}\big(\text{softmax}(z_i^{\text{old}}/T) \,\|\, \text{softmax}(z_i/T)\big), \tag{22}$$

where $z_i^{\text{old}} = g^{\text{old}}(\tilde{\mathbf{h}}_i)$ and $T \in [2, 5]$ is the temperature. A larger $T$ smooths distributions and highlights relative class probabilities (Hinton et al., 2015). It also improves probability calibration (Guo et al., 2017). We set $T = 3$, which balances stability and informativeness.

The second term is *anchor distillation* on slot anchors $\mathcal{A}$:

$$\mathcal{L}_{\text{anchor}} = \frac{T^2}{|\mathcal{A}|} \sum_{a \in \mathcal{A}} \text{KL}\big(\text{softmax}(z_a^{\text{old}}/T) \,\|\, \text{softmax}(z_a/T)\big), \tag{23}$$

where $z_a^{\text{old}} = g^{\text{old}}(\mathbf{a})$ and $z_a = g(\mathbf{a})$. Anchors are surrogate prototypes maintained by SGSE. They store old knowledge without raw replay and follow the idea of prototype learning (Snell et al., 2017).

Therefore, the full Stage 2 objective is

$$\mathcal{L}^{(2)} = \mathcal{L}_{\text{CE}} + \lambda_{\text{LwF}} \mathcal{L}_{\text{LwF}} + \lambda_{\text{anchor}} \mathcal{L}_{\text{anchor}}, \tag{24}$$

with $\lambda_{\text{LwF}} \approx 1.0$ and $\lambda_{\text{anchor}} \in [0.5, 1.0]$. These weights reflect the balance between plasticity (cross-entropy) and stability (distillation). Anchor distillation connects SGSE anchors with the following theoretical bound:

**Theorem 2.** *Assume: (i) $g, g^{\text{old}} : \mathbb{R}^{d_T} \to \mathbb{R}^C$ are L-Lipschitz in logits, (ii) for all anchors $a \in \mathcal{A}$, $\text{KL}\big(\text{softmax}(g^{\text{old}}(a)/T) \,\|\, \text{softmax}(g(a)/T)\big) \leq \eta$, and (iii) every old-class feature $\tilde{\mathbf{h}}$ lies within distance $\delta$ of some anchor $a$ in feature space. Then for any such $\tilde{\mathbf{h}}$,*

$$\big\|\text{softmax}(g^{\text{old}}(\tilde{\mathbf{h}})/T) - \text{softmax}(g(\tilde{\mathbf{h}})/T)\big\|_1 = O\Big(\sqrt{\eta} + \frac{L}{T}\delta\Big), \tag{25}$$

*and the old-class accuracy drop is $O\big(\sqrt{\eta} + L\delta/T\big)$.*

*Proof.* By (ii) and Pinsker's inequality (Canonne, 2022), the softmax distributions at each anchor differ by at most $O(\sqrt{\eta})$ in $\ell_1$. By (i), logits vary at most $L\delta$ within a $\delta$-ball. After temperature scaling, this variation adds at most $O((L/T)\delta)$ in probability space. By the triangle inequality, the total deviation is $O(\sqrt{\eta} + (L/T)\delta)$, which yields the stated bound. □

This theorem shows that anchor-based distillation gives provable retention. If anchors approximate old features within $\delta$, and if distillation keeps anchor predictions consistent within $\eta$, then the deviation on old-class predictions is tightly bounded. Thus, DCDA preserves knowledge without raw replay and remains both memory-efficient and theoretically sound.

**Takeaways 3.2.** MARS *avoids raw replay by encoding knowledge into slots and anchors. SGSE enables principled slot growth, and DCDA integrates new capacity through contrastive alignment and anchor-based distillation. With the encoder frozen, adaptation remains efficient. Empirically (Sec. 4), DCDA improves accuracy by up to $20\%$ relative to DER++ (Buzzega et al., 2020), depending on the dataset. Together, SGSE and DCDA offer a principled solution to the stability–plasticity tradeoff in continual learning with large pre-trained models.*

## 3.3 COMPUTE AND MEMORY COMPLEXITY

At last, we analyze the computational and storage costs of MARS and show how SGSE keeps growth both controlled and predictable.

**Per-example Overhead.** Each forward pass consists of the frozen encoder $f(\cdot)$, followed by the memory router and the slot-conditioned affine transform. Routing costs $O(S_t d_k)$ per input because it computes query–key similarities, and affine adaptation costs $O(S_t d_T)$. Thus the per-example overhead is

$$\text{Time}(x_t) = O\big(S_t(d_k+d_T)\big) \;=\; O(S_t d_T) \quad \text{if } d_k \leq d_T. \tag{26}$$

Training is efficient because Stage 1 updates only $(W_q, K, \boldsymbol{\gamma}, \boldsymbol{\beta})$ and Stage 2 updates only $g$, both of which are much smaller than the backbone.

**Per-slot Cost.** Each slot stores a key $k_i \in \mathbb{R}^{d_k}$, affine parameters $(\boldsymbol{\gamma}_i, \boldsymbol{\beta}_i) \in \mathbb{R}^{2d_T}$, and an anchor $\mathbf{a}_i \in \mathbb{R}^{d_T}$. This amounts to $O(d_k+d_T)$ parameters per slot, plus the head $|g|$. During inference, routing and adaptation scale linearly with $S_t$ and remain independent of the frozen encoder $|f|$.

**Lemma 1.** *With $S_t$ slots and feature dimension $d_T$, the per-input compute cost is*

$$O\big(S_t(d_k+d_T)\big) \quad (\text{reducing to } O(S_t d_T) \text{ if } d_k \leq d_T), \tag{27}$$

*and the parameter footprint is*

$$O\big(S_t(d_k+d_T)\big) + |g|. \tag{28}$$

**Complexity Control via SGSE.** Without regulation, $S_t$ could grow linearly with stream length $T$, leading to uncontrolled complexity. SGSE avoids this by allowing slot expansion only when there is statistically significant evidence that existing slots cannot cover new inputs. This mechanism ensures that growth is linked to true novelty rather than noise. Formally, Cor. 1 shows that the false-expansion probability per test is at most $\alpha$, which provides a bound on the expected growth:

**Proposition 2.** *For SGSE with Wilson test level $\alpha$, evaluated every $m$ samples over a window $n \geq m$, let $T$ be the stream length, $M = \lfloor (T - w)/m \rfloor$ the number of tests, and $S_T$ the slot count at horizon $T$. Then*

$$\mathbb{E}[S_T] \;\leq\; S_0 + N_T + \alpha M, \tag{29}$$

*where $N_T$ is the number of true novelty expansions. Moreover, with probability $\geq 1 - \delta$,*

$$S_T \;\leq\; S_0 + N_T + \alpha M + \sqrt{\tfrac{M}{2} \ln \tfrac{1}{\delta}}. \tag{30}$$

**Theorem 3.** *Combining Lemma 1 and Prop. 2, the expected per-example cost at time $T$ is*

$$\mathbb{E}[\text{Time}(x_T)] = O\Big((d_k+d_T)\big(S_0 + \mathbb{E}[N_T] + \alpha M\big)\Big), \tag{31}$$

*with a high-probability bound of the same form. The parameter footprint satisfies*

$$\mathbb{E}[\text{Mem}_T] = O\Big((d_k+d_T)\big(S_0 + \mathbb{E}[N_T] + \alpha M\big)\Big) + |g|. \tag{32}$$

**Takeaways 3.3.** *When the number of true novelties $N_T$ grows sublinearly with $T$ (for example $O(\log T)$ or $O(T^\rho)$ with $\rho < 1$), both computation and memory also grow sublinearly, while scaling linearly with $d_T$ and $S_t$. In this case, MARS scales smoothly with streaming data and avoids uncontrolled overhead. In contrast, heuristic expansion methods often cause unbounded slot growth and lead to linear or even superlinear complexity. By grounding expansion in SGSE's statistical test, MARS provides controlled growth with both efficiency and scalability.*

## 4 EXPERIMENTS

### 4.1 EXPERIMENTAL SETUP

**Datasets and Metric.** We evaluate MARS on both vision and NLP tasks using standard benchmarks. For vision tasks, we adopt CIFAR-100 (Krizhevsky & Hinton, 2009), which contains 100 classes with 50,000 training images and 10,000 test images of size $32 \times 32$, and Tiny-ImageNet (Le & Yang, 2015), which includes 200 classes with 500 training, 50 validation, and 50 test images per class of size $64 \times 64$. Following standard class-incremental protocols (Han & Guo, 2022; Liu et al., 2024; Pietron et al., 2025), CIFAR-100 is divided into 10 tasks with 10 classes each, and Tiny-ImageNet into 10 tasks with 20 classes each. For NLP tasks, we use 19 aspect-based sentiment classification (ASC) datasets adopted in prior work (Ke et al., 2021b), where each dataset corresponds to a product domain such as laptops, restaurants, cameras, or phones, and is annotated with three sentiment polarities: positive, neutral, and negative. Each dataset is treated as one task, which enables evaluation of MARS under diverse domains, different class sizes, and distribution shifts. After training on task $t$, the model is evaluated on the test sets of all tasks $1, \dots, t$, and the average accuracy $\bar{A}_t = \frac{1}{t} \sum_{i=1}^{t} a_{t,i}$ is computed, where $a_{t,i}$ is the accuracy on task $i$ after learning task $t$. This produces a trajectory of average accuracy as tasks accumulate, which typically decreases due to forgetting. Unless otherwise noted, we report $\bar{A}_T$, the average accuracy after completing the entire sequence. All experiments are conducted with random seeds $\{12, 123, 1234\}$ on NVIDIA RTX 5090 GPUs.

**Baselines and Settings.** We compare MARS with representative continual learning methods, including EWC (Kirkpatrick et al., 2017), iCaRL (Rebuffi et al., 2017), DER++ (Buzzega et al., 2020), LDC (Gomez-Villa et al., 2024), and PASS++ (Zhu et al., 2025). To ensure fairness, each baseline is evaluated under two settings. In the standard setting, the entire backbone is trainable as in the original method. In the frozen-encoder setting, the backbone is fixed and only lightweight components such as task-specific heads or adapters are updated. This matches the capacity used by MARS and avoids bias toward methods that gain mainly from updating a large number of backbone parameters. Replay-based methods (iCaRL, DER++, PASS++) are restricted to an exemplar budget comparable to the anchor storage in MARS. In addition, all methods use the same encoder, training schedule, and evaluation protocol to ensure consistent comparisons.

**Implementation Details.** For vision benchmarks, we use CLIP (Radford et al., 2021) as the frozen encoder $f(\cdot)$, with its vision transformer (ViT-B/16) producing features of dimension $d_T$. For NLP tasks, we use BERT-base (Devlin et al., 2019), also with frozen parameters. On top of the encoder, the memory router is implemented as a linear projection $W_q$ that maps frozen features into a query space of dimension $d_k = 64$, which is then compared with the slot key set $K = \{k_i\}_{i=1}^{S_t}$ to compute routing probabilities. We initialize with $S_0 = 32$ slots, set the quantile momentum to $\beta = 0.9$, and adopt a Wilson score threshold at 95% confidence. Training follows the two-stage DCDA protocol. In Stage 1 (feature adaptation), we update only the memory parameters $(W_q, K, \boldsymbol{\gamma}, \boldsymbol{\beta})$ for 20 epochs using supervised contrastive loss with batch size 128 and temperature $\tau = 0.07$, together with a smoothness tether weighted by $\lambda_{\text{smooth}} = 0.3$. In Stage 2 (classifier tuning), we fix the memory and train the classifier $g$ for 20 epochs with cross-entropy loss and two distillation terms. The learning rate is 0.001, and entropy regularization is optionally applied with coefficient 0.1.

### 4.2 EXPERIMENTAL RESULTS

**Main Results.** Table 1 reports the average accuracy across benchmarks. Replay-based methods such as DER++ and PASS++ outperform regularization-based methods such as EWC, but their reliance on small exemplar memories causes performance to plateau as the task sequence increases. On CIFAR-100 and Tiny-ImageNet, these methods converge around 52–54%, while MARS consistently achieves 56–58%, a relative gain of about 3–5%. On ASC, DER++ and PASS++ stabilize near 74–

Table 1: Average accuracy of different methods under standard and frozen-encoder settings.

| Algorithm | CIFAR-100 | | Tiny-ImageNet | | ASC | |
|---|---|---|---|---|---|---|
| | Standard | Frozen | Standard | Frozen | Standard | Frozen |
| **Fine-tune** | 30.74±0.43 | 30.26±0.20 | 28.32±0.65 | 28.27±0.43 | 60.90±0.29 | 61.30±0.80 |
| **EWC** | 47.84±0.58 | 47.60±0.40 | 36.47±0.54 | 36.38±0.39 | 70.26±0.66 | 70.66±0.69 |
| **DER++** | 52.24±0.66 | 51.72±0.47 | 40.99±0.37 | 40.87±0.16 | 75.53±0.27 | 75.91±0.21 |
| **LDC** | 54.14±0.17 | 53.95±0.48 | 43.39±0.63 | 43.41±0.55 | 75.11±0.60 | 75.49±0.23 |
| **PASS++** | 53.67±0.50 | 52.92±0.52 | 42.31±0.61 | 42.53±0.70 | 74.72±0.20 | 75.22±0.73 |
| **ours** | 57.33±0.48 | 57.50±0.54 | 49.12±0.36 | 49.46±0.14 | 79.45±0.25 | 79.85±0.66 |

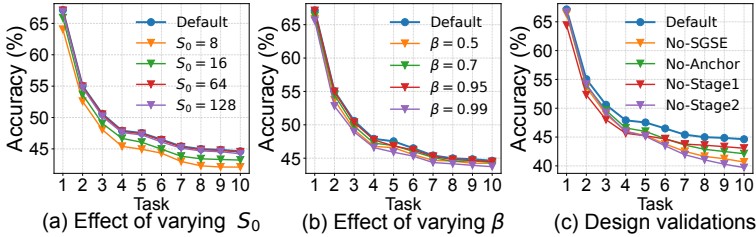

(a) Effect of varying $S_0$    (b) Effect of varying $\beta$    (c) Design validations

Figure 2: Ablation study results on Tiny-ImageNet.

75%, whereas MARS reaches 78–79%, showing that it retains domain-specific knowledge without raw data. LDC also improves over DER, but its gains remain below those of MARS, suggesting that heuristic consolidation is less effective than statistically grounded slot expansion with anchor distillation. Another important observation is that the difference between the standard and frozen-encoder settings is usually within 1–2%, rather than a fixed gap. This shows that improvements do not come from updating the backbone, but from how models allocate and preserve capacity for new tasks. By combining statistical slot expansion with dual-stage adaptation, MARS achieves a better balance between stability and plasticity. Through controlled expansion and anchor-based retention, it consistently provides higher accuracy under the same memory budget, demonstrating its suitability for continual learning with large pre-trained models.

**Effect of Varying $S_0$.** Figure 2(a) shows that the initial slot number $S_0$ strongly influences performance. A small $S_0$ (e.g., $S_0=8$) causes accuracy to drop quickly after a few tasks due to limited capacity and strong interference. Increasing $S_0$ to 16–64 improves performance, with the best results at $S_0=32$, which maintains higher accuracy across tasks. Enlarging $S_0$ to 128 gives no benefit and slightly degrades later accuracy, likely from redundant slots and noisy routing. These results confirm that initialization is important: too few slots reduce plasticity, while too many reduce stability.

**Effect of Varying $\beta$.** Figure 2(b) analyzes the smoothing coefficient $\beta$, which controls how the statistical trigger adapts to shifts in routing confidence. A small $\beta$ (e.g., 0.5) causes unstable quantile estimates, leading to premature expansions and lower accuracy. As $\beta$ increases to 0.7–0.95, performance improves steadily, with $\beta=0.9$ offering the most robust balance. When $\beta$ is too large (0.99), the estimator reacts too slowly to distributional shifts, delaying necessary expansions and harming late-task accuracy. These findings validate our choice of $\beta=0.9$, which balances stability and responsiveness for continual learning.

**Validation of Design.** Figure 2(c) highlights the complementary roles of SGSE, anchors, and the two-stage adaptation. Removing SGSE leads to a steep accuracy drop (final accuracy ∼41%), confirming that statistically grounded slot expansion is essential for maintaining sufficient capacity. Removing anchors causes a similar decline (final accuracy ∼42%), underscoring their importance for knowledge retention without replay. Disabling Stage 1 (contrastive feature adaptation) reduces representation alignment (final ∼43%), while omitting Stage 2 (classifier distillation) yields the lowest accuracy (final ∼40%), showing that both stages are necessary. Together, these results show that SGSE, anchors, and dual-stage adaptation work together: SGSE regulates expansion, anchors preserve knowledge, and dual-stage adaptation balances stability and plasticity.

**Anchor Diagnostics.** We further examine the behaviour of the anchor space using three empirical diagnostics. Since anchors and routed features lie in the same feature space $\mathbb{R}^{d_T}$, cosine similarity provides a direct way to assess how each anchor relates to the features assigned to its slot. Across tasks, these similarity values remain within the range 0.60–0.85 and vary smoothly as new classes are introduced. To assess temporal stability, we compare each anchor to its counterpart after consecutive

tasks and obtain stability scores between 0.65 and 0.98, indicating that the updates are gradual rather than abrupt. A nearest-neighbor inspection further shows that anchors tend to remain associated with coherent groups of feature patterns, such as vehicles, animals, or background textures. Together, these diagnostics suggest that the anchor space preserves a stable and interpretable structure throughout the task sequence. Additional analyses are provided in Appendix A.4.

**Slot Growth.** We visualize how the number of slots changes during training on Tiny-ImageNet, CIFAR-100, and ASC in Figure 3. In all cases, SGSE expands the memory only when the confidence statistic exceeds the Wilson bound for several steps. The slot count grows steadily during the early tasks and then approaches a stable value as learning continues. On Tiny-ImageNet, the slot count increases from $S_0 = 32$ to about $S_T = 49$. On CIFAR-100, it reaches approximately $S_T = 44$. On ASC, it increases to around $S_T = 58$ as more domains are introduced. These results are consistent with the theoretical analysis and show that SGSE provides smooth and controlled capacity expansion.

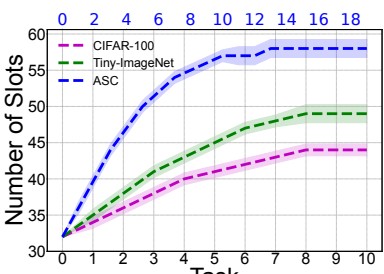

Figure 3: Slot growth across tasks.

**Extended Baseline Comparisons.** We include additional PTM and PEFT baselines under the same frozen-encoder protocol. These baselines include L2P (Wang et al., 2022), CODA-Prompt (Smith et al., 2023), and a representative CLIP-oriented method (Jha et al., 2024). All methods use the frozen CLIP ViT-B/16 backbone and have train-

Table 2: Extended baseline comparisons.

| Method | CIFAR-100 | Tiny-IN | ASC |
|---|---|---|---|
| L2P | 52.30±0.45 | 43.80±0.32 | 73.90±0.51 |
| CODA-Prompt | 54.71±0.61 | 45.10±0.58 | 74.70±0.44 |
| CLAP4CLIP | 55.42±0.50 | 46.50±0.64 | – |
| MARS | **57.50±0.54** | **49.46±0.14** | **79.85±0.66** |

able components on the order of $10^6$ parameters, ensuring comparable effective capacity. Across all benchmarks, MARS achieves the highest accuracy. These results show that the gains of MARS come from statistical slot expansion and anchor-based distillation rather than from prompting strategies.

**Scalability Analysis.** We also evaluate the method on ImageNet-100. MARS reaches 49.46% on Tiny-ImageNet, which is 2.96 points higher than the CLIP-oriented baseline. On ImageNet-100, MARS also performs better than the best frozen-

Table 3: Performance on larger-scale data.

| Dataset | Best Baseline | MARS | Final $S_T$ |
|---|---|---|---|
| Tiny-ImageNet | 46.50±0.64 | 49.46±0.14 | $\approx 49$ |
| ImageNet-100 | 39.67±0.60 | 42.08±0.53 | $\approx 65$ |

backbone baseline. During this evaluation, the slot count grows from $S_0 = 32$ to about $S_T = 65$. This growth remains moderate and shows that SGSE maintains stable and predictable capacity expansion as the dataset size and complexity increase.

**Parameter and Inference Cost.** We compare parameter count and inference time on Tiny-ImageNet with PTM/PEFT baselines under the frozen-encoder setting (L2P, CODA-Prompt, CLAP4CLIP). These baselines typically use 0.5M–0.8M trainable parameters, whereas MARS re-

Table 4: Parameter and inference cost.

| Metric | Baselines | MARS |
|---|---|---|
| Trainable parameters | 0.5M to 0.8M | 0.2M |
| Inference time per batch | 7.8ms to 8.1ms | 8.5ms |
| Final accuracy (%) | 43.80 to 46.50 | **49.46** |

quires only 0.2M, making it substantially lighter. Despite dynamic expansion, the inference overhead remains small: MARS reaches 8.5ms per batch, only a minor increase over the baselines' 7.8–8.1ms. Within this group of methods, accuracy ranges from 43.8% to 46.5%, while MARS achieves 49.46%.

## 5 CONCLUSIONS AND LIMITATIONS

In conclusion, we present the MARS framework for continual learning with large pre-trained models, which integrates statistical slot expansion, anchor-based retention, and a dual-stage adaptation strategy. This design improves the stability–plasticity balance while remaining scalable under practical constraints. A key advantage is its reliance on frozen encoders and lightweight modules, making it applicable to both vision and language tasks. Despite these strengths, the framework has limitations. It depends on a reliable frozen encoder, which may not capture fine-grained features in new domains. It also requires careful tuning of hyperparameters that control expansion and adaptation. In addition, although the method reduces reliance on raw data, it does not remove memory costs entirely. Addressing these challenges is an important direction for future work.

ACKNOWLEDGMENTS

This work was fully supported by the National Natural Science Foundation of China for Excellent Young Scientists Fund Program (Overseas), awarded in 2024.

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

## A  APPENDIX

### A.1  THE USE OF LARGE LANGUAGE MODELS (LLMs)

During the preparation of this paper, we made limited use of large language models as writing assistants. Their role was restricted to checking grammar, improving clarity, and polishing exposition. All technical ideas, methods, and experiments were fully developed and validated by the authors.

### A.2  DETAILS OF THEORETICAL FOUNDATION

**Proposition 1 (Monotonicity of $s_t$ in $c_t$).**  This result shows that the top-slot confidence $s_t = \max_i p_i(\mathbf{x}_t)$ behaves as a calibrated statistic for novelty detection: when the similarity between the query and its best-matching key increases, the corresponding softmax confidence increases strictly, provided the other similarities are fixed.

*Proof.* Let $a_j = \langle \hat{q}(\mathbf{x}_t), \hat{k}_j \rangle$, and let $i^\star \in \arg\max_j a_j$ with $c := a_{i^\star}$. All other similarities $\{a_j\}_{j \neq i^\star}$ are treated as constants during this analysis. Define

$$A := \sum_{j \neq i^\star} e^{a_j/\tau_r}, \qquad A > 0 \text{ since } S_t > 1.$$

Then the maximum softmax confidence is

$$s(c) = \frac{e^{c/\tau_r}}{e^{c/\tau_r} + A} = \frac{1}{1 + Ae^{-c/\tau_r}}.$$

This is a logistic-type function of $c$, strictly between 0 and 1. Differentiating with respect to $c$ gives

$$\frac{ds}{dc} = \frac{1}{\tau_r} \frac{Ae^{-c/\tau_r}}{(1 + Ae^{-c/\tau_r})^2} = \frac{1}{\tau_r} s(c)\big(1 - s(c)\big).$$

Since $\tau_r > 0$ and $0 < s(c) < 1$, the derivative is positive. Thus, conditional on other similarities being fixed, the top-slot confidence $s_t$ is strictly increasing in $c_t$, i.e. the maximum cosine similarity. This monotonicity means $s_t$ faithfully reflects changes in slot affinity, making it a suitable indicator.  $\square$

**Theorem 1 (EMA quantile tracker under weak dependence).** This result analyzes the exponentially smoothed quantile statistic $Q_t$ that underlies SGSE. We show (i) convergence in mean square to the long-run quantile and (ii) a predictable timescale for detection after a mean shift.

*Proof.* As defined by:
$$Q_t = \beta Q_{t-1} + (1-\beta)q_t, \qquad \beta \in [0,1),$$
where $\{q_t\}$ is a stationary sequence with $\mathbb{E}[q_t] = q^\star$. For clarity, first assume $\{q_t\}$ are i.i.d. with variance $\sigma_q^2$. By taking expectations, we have
$$\mathbb{E}[Q_t] = \beta \, \mathbb{E}[Q_{t-1}] + (1-\beta)q^\star.$$
This is a standard linear recursion with solution
$$\mathbb{E}[Q_t] = q^\star + \beta^t(Q_0 - q^\star).$$
Hence $Q_t$ converges in expectation to $q^\star$ as $t \to \infty$. Then, for the variance,
$$\mathrm{Var}(Q_t) = \beta^2 \mathrm{Var}(Q_{t-1}) + (1-\beta)^2\sigma_q^2.$$
Unrolling this recursion,
$$\mathrm{Var}(Q_t) = (1-\beta)^2\sigma_q^2 \sum_{i=0}^{t-1} \beta^{2i} = \frac{(1-\beta)^2}{1-\beta^2}\sigma_q^2(1-\beta^{2t}).$$
As $t \to \infty$, this converges to $\frac{(1-\beta)^2}{1-\beta^2}\sigma_q^2$. Thus $Q_t \to q^\star$ in $L^2$. If $q_t$ are not i.i.d. but weakly dependent (e.g., $\alpha$-mixing), the same result holds with $\sigma_q^2$ replaced by the long-run variance. Further, suppose at time $\tau$ the mean shifts from $q^\star$ to $q' < q^\star$. For $k \geq 0$,
$$\mathbb{E}[Q_{\tau+k}] = q' + \beta^k(\mathbb{E}[Q_\tau] - q').$$
Fix a threshold $\theta$ with $q' < \theta < q^\star$. The smallest integer $k$ such that $\mathbb{E}[Q_{\tau+k}] \leq \theta$ must satisfy
$$\beta^k \leq \frac{\theta - q'}{\mathbb{E}[Q_\tau] - q'}.$$
Taking logarithms,
$$k \geq \frac{\ln\left(\frac{\mathbb{E}[Q_\tau] - q'}{\theta - q'}\right)}{-\ln \beta}.$$
Using the inequality $-\ln \beta \geq 1-\beta$ for $\beta \in [0,1)$, we obtain
$$k \leq \frac{1}{1-\beta}\ln\left(\frac{\mathbb{E}[Q_\tau] - q'}{\theta - q'}\right).$$
Therefore, the *mean-crossing index* (i.e., how many steps until the expected trajectory falls below $\theta$) scales as $O((1-\beta)^{-1})$. This provides a predictable detection timescale: smaller $(1-\beta)$ (i.e., heavier smoothing) leads to slower adaptation. $\square$

**Corollary 1 (False expansion control).** This establishes that the Wilson lower-bound test provides approximate per-test false expansion control at level $\alpha$.

*Proof.* Let $X_1, \ldots, X_n \sim$ i.i.d. Bernoulli$(p)$ with $p = \Pr(s_t \geq Q_t) \geq \tau$. Define $\hat{p}_n = \frac{1}{n}\sum_{i=1}^n X_i$. The one-sided Wilson lower bound $\mathrm{LB}(\hat{p}_n; n, z)$ with $z = \Phi^{-1}(1-\alpha)$ satisfies, by score-test theory,
$$\Pr\big(\mathrm{LB}(\hat{p}_n; n, z) \leq p\big) \geq 1-\alpha.$$
Since $p \geq \tau$, the event $\{\mathrm{LB} < \tau\}$ implies $\{\mathrm{LB} < p\}$. Therefore,
$$\Pr(\mathrm{LB}(\hat{p}_n; n, z) < \tau) \leq \Pr(\mathrm{LB}(\hat{p}_n; n, z) < p) \leq \alpha,$$
up to normal approximation error. Thus the per-test false expansion probability is approximately controlled at level $\alpha$. $\square$

**Theorem 2 (Anchor-based retention).** This theorem shows that, under mild assumptions, anchor-based distillation guarantees bounded deviation between the old and new models' predictions on old-class features.

*Proof.* Let $p(u) = \mathrm{softmax}(u/T)$ denote the temperature-scaled softmax. By assumption (ii), for each anchor $a \in \mathcal{A}$,

$$\mathrm{KL}\big(p(g^{\mathrm{old}}(a)) \,\|\, p(g(a))\big) \leq \eta.$$

By Pinsker's inequality,

$$\|p(g^{\mathrm{old}}(a)) - p(g(a))\|_1 \leq \sqrt{2\eta}.$$

Now consider any old-class feature $\tilde{\mathbf{h}}$ within distance $\delta$ of some anchor $a$. By Lipschitz continuity of logits (assumption (i)),

$$\|g(\tilde{\mathbf{h}}) - g(a)\|_2 \leq L\delta, \qquad \|g^{\mathrm{old}}(\tilde{\mathbf{h}}) - g^{\mathrm{old}}(a)\|_2 \leq L\delta.$$

And the Jacobian of $p(u)$ is

$$\nabla p(u) = \frac{1}{T}\big[\mathrm{Diag}(p(u)) - p(u)p(u)^\top\big].$$

Its operator norm is bounded by $1/(2T)$ in $\ell_2 \to \ell_2$ norm. Thus, by the mean-value theorem,

$$\|p(g(\tilde{\mathbf{h}})) - p(g(a))\|_1 \leq \sqrt{C} \cdot \|\nabla p(\xi)\|_{2 \to 2} \cdot \|g(\tilde{\mathbf{h}}) - g(a)\|_2 \leq \frac{\sqrt{C}}{2T} L\delta,$$

and similarly

$$\|p(g^{\mathrm{old}}(\tilde{\mathbf{h}})) - p(g^{\mathrm{old}}(a))\|_1 \leq \frac{\sqrt{C}}{2T} L\delta.$$

Here $\sqrt{C}$ comes from $\|v\|_1 \leq \sqrt{C}\|v\|_2$, and can be absorbed into big-$O$ notation. By applying the triangle inequality, we have

$$\|p(g^{\mathrm{old}}(\tilde{\mathbf{h}})) - p(g(\tilde{\mathbf{h}}))\|_1 \leq \|p(g^{\mathrm{old}}(a)) - p(g(a))\|_1 + \tfrac{L}{T}\delta \leq \sqrt{2\eta} + \tfrac{L}{T}\delta \cdot O(1).$$

Hence, for any old-class feature, the deviation between old and new softened predictions is bounded by $O(\sqrt{\eta} + (L/T)\delta)$. Under mild posterior-margin conditions, this ensures the drop in classification accuracy is controlled at the same order. $\square$

**Proposition 2 (Slot growth bound).** This proposition shows that SGSE separates true expansions (driven by genuine novelty) from false expansions (caused by noise), and that the latter are statistically controlled.

*Proof.* Let $M = \lfloor (T - w)/m \rfloor$ denote the number of hypothesis tests up to time $T$. For each test $j$, let $Y_j \in \{0, 1\}$ be the indicator of a false expansion. By Corollary 1,

$$\Pr(Y_j = 1) \leq \alpha.$$

Thus

$$\mathbb{E}[Y_j] \leq \alpha, \qquad \mathbb{E}[F] \leq \alpha M, \quad \text{where } F = \sum_{j=1}^{M} Y_j.$$

If we ensure test windows are disjoint (i.e., $n \leq m$), then the $Y_j$'s are independent. By Hoeffding's inequality,

$$\Pr(F - \mathbb{E}[F] \geq \epsilon) \leq \exp\left(-\frac{2\epsilon^2}{M}\right).$$

Choosing $\epsilon = \sqrt{\frac{M}{2} \ln(1/\delta)}$ yields

$$F \leq \alpha M + \sqrt{\tfrac{M}{2} \ln \tfrac{1}{\delta}}, \qquad \text{with prob. } \geq 1 - \delta.$$

Let $N_T$ be the number of true expansions. Then the total slot count is

$$S_T \leq S_0 + N_T + F.$$

Taking expectations,

$$\mathbb{E}[S_T] \leq S_0 + \mathbb{E}[N_T] + \alpha M,$$

and the high-probability bound follows from the inequality above. $\square$

---

**Algorithm 1** MARS: Training with SGSE and DCDA

---

1: **for** each task $t = 1, \ldots, T$ **do**
2:     Initialize buffers: success buffer $\mathcal{B}$ (size $n$) and low-confidence buffer $\mathcal{L}$.
3:     Initialize quantile tracker $Q_{t,0}$ with the first batch.
4:     **Stage 1: Feature Adaptation (memory-only)**
5:     **for** each mini-batch $\mathcal{D}_t$ **do**
6:         Extract frozen features $\mathbf{h}_T \leftarrow f(\mathbf{x})$ and queries $q \leftarrow W_q \mathbf{h}_T$, $\hat{q} \leftarrow q/\|q\|$.
7:         Compute routing probabilities $p_i(\mathbf{x}) \propto \exp(\langle \hat{q}, \hat{k}_i \rangle / \tau_r)$ and top confidence $s(\mathbf{x})$.
8:         Update slot statistics $\mu_i, c_i$ with EMA ($\alpha = 0.05$) and anchors $\mathbf{a}_i$.
9:         Update quantile $q_t$ from last $w$ samples and smooth $Q_t \leftarrow \beta Q_{t-1} + (1 - \beta) q_t$.
10:        Record Bernoulli trial $X(\mathbf{x})$ in buffer $\mathcal{B}$; compute empirical success rate $\hat{p}$.
11:        **if** $\mathrm{LB}(\hat{p}; n, z) < \tau_{\mathrm{succ}}$ (Wilson lower bound test) **then**
12:            **Expand:** Add new slot $j$ with key $k_j$ from mean query of $\mathcal{L}$; set $(\boldsymbol{\gamma}_j, \boldsymbol{\beta}_j) = (\mathbf{1}, \mathbf{0})$
13:        **end if**
14:        Update $\mathcal{L}$ with lowest-confidence samples in batch.
15:        Compute adapted features $\tilde{\mathbf{h}} = (\sum_i p_i \boldsymbol{\gamma}_i) \odot \mathrm{LN}(\mathbf{h}_T) + (\sum_i p_i \boldsymbol{\beta}_i)$.
16:        Optimize memory by minimizing $\mathcal{L}^{(1)} = \mathcal{L}_{\mathrm{supcon}}(\tilde{\mathbf{h}}; \tau) + \lambda_{\mathrm{smooth}} \|\tilde{\mathbf{h}} - \mathbf{h}_T\|_2^2$.
17:     **end for**
18:     **Stage 2: Classifier Tuning (head-only)**
19:     Store old classifier $g^{\mathrm{old}} \leftarrow g$.
20:     **for** each mini-batch $\mathcal{D}_t$ **do**
21:         Compute logits $z \leftarrow g(\tilde{\mathbf{h}})$, $z^{\mathrm{old}} \leftarrow g^{\mathrm{old}}(\tilde{\mathbf{h}})$.
22:         Compute anchor logits $z_a \leftarrow g(\mathbf{a})$, $z_a^{\mathrm{old}} \leftarrow g^{\mathrm{old}}(\mathbf{a})$ for $a \in \mathcal{A}$.
23:         Minimize $\mathcal{L}^{(2)}$ and update only $g$.
24:     **end for**
25: **end for**
26: **return** $(W_q, K, \boldsymbol{\gamma}, \boldsymbol{\beta}, g)$.

---

**Theorem 3 (Overall complexity).** Finally, we connect slot growth to computational and memory costs.

*Proof.* From Lemma 1,

$$\mathrm{Time}(x_t) = \Theta\big((d_k + d_T)S_t\big), \qquad \mathrm{Mem}_t = \Theta\big((d_k + d_T)S_t\big) + |g|.$$

Taking expectations and substituting Proposition 2,

$$\mathbb{E}[\mathrm{Time}(x_T)] = O\Big((d_k + d_T)\big(S_0 + \mathbb{E}[N_T] + \alpha M\big)\Big),$$

$$\mathbb{E}[\mathrm{Mem}_T] = O\Big((d_k + d_T)\big(S_0 + \mathbb{E}[N_T] + \alpha M\big)\Big) + |g|.$$

For the high-probability bound, we replace $S_T$ by its probabilistic upper bound in Proposition 2, which yields the same asymptotic order. Thus both compute and memory scale linearly with slot count, and slot count itself is controlled by SGSE. $\qquad\square$

### A.3 OVERALL WORKFLOW OF MARS

The overall design of MARS integrates two complementary mechanisms on top of the frozen LPM backbone. As shown in Algorithm 1, *SGSE* monitors router confidences and decides when to create new slots by formulating expansion as a statistical decision problem with guarantees on false alarms and detection delay. When a new slot is added, *DCDA* controls its integration: Stage 1 aligns slot features through supervised contrastive learning with smoothness regularization, and Stage 2 tunes the classifier with Learning-without-Forgetting distillation on current inputs and anchor-based distillation on surrogate prototypes. This workflow ensures controlled slot growth, efficient adaptation, and a provable stability–plasticity balance without updating the large pre-trained encoder.

Table 5: Anchor–feature similarity on Tiny-ImageNet.

| Anchor | After Task 1 | After Task 2 | After Task 3 | After Task 4 | After Task 5 |
|--------|--------------|--------------|--------------|--------------|--------------|
| A1 | $0.782 \pm 0.046$ | $0.759 \pm 0.038$ | $0.746 \pm 0.041$ | $0.762 \pm 0.029$ | $0.755 \pm 0.040$ |
| A2 | $0.842 \pm 0.049$ | $0.825 \pm 0.050$ | $0.807 \pm 0.042$ | $0.792 \pm 0.034$ | $0.781 \pm 0.056$ |
| A3 | $0.603 \pm 0.029$ | $0.618 \pm 0.033$ | $0.635 \pm 0.042$ | $0.648 \pm 0.047$ | $0.662 \pm 0.039$ |
| A4 | $0.701 \pm 0.027$ | $0.718 \pm 0.038$ | $0.734 \pm 0.041$ | $0.725 \pm 0.039$ | $0.712 \pm 0.026$ |
| **Anchor** | **After Task 6** | **After Task 7** | **After Task 8** | **After Task 9** | **After Task 10** |
| A1 | $0.770 \pm 0.042$ | $0.758 \pm 0.040$ | $0.749 \pm 0.045$ | $0.761 \pm 0.043$ | $0.752 \pm 0.036$ |
| A2 | $0.794 \pm 0.027$ | $0.786 \pm 0.055$ | $0.778 \pm 0.061$ | $0.791 \pm 0.034$ | $0.783 \pm 0.048$ |
| A3 | $0.671 \pm 0.041$ | $0.658 \pm 0.041$ | $0.645 \pm 0.022$ | $0.661 \pm 0.044$ | $0.653 \pm 0.051$ |
| A4 | $0.728 \pm 0.026$ | $0.735 \pm 0.053$ | $0.742 \pm 0.049$ | $0.726 \pm 0.046$ | $0.732 \pm 0.037$ |

Table 7: Nearest neighbor classes for selected anchors on Tiny-ImageNet.

| Anchor | Nearest classes | Interpretation |
|--------|-----------------|----------------|
| A1 | truck, ship, bus, related vehicle classes | rigid objects or vehicles |
| A2 | dog, cat, deer, bird | animal categories |
| A3 | classes with frequent sky or water textures* | textures or background |
| A4 | bird, airplane, ship | open or airborne scenes |

## A.4 ANCHOR COVERAGE DIAGNOSTICS

This section provides additional diagnostics that examine the coverage assumption used in Theorem 2. We evaluate the behaviour of the anchors on Tiny-ImageNet under the frozen-encoder setting. Because anchors and routed features share the same feature space $\mathbb{R}^{d_T}$, we can compare them directly using cosine similarity. We report three diagnostics that characterize anchor–feature similarity, temporal stability, and semantic coherence.

**Anchor–feature Similarity.** We first study how each anchor relates to the routed features assigned to its slot. We randomly sample four anchors and compute the cosine similarity between each anchor and the router-weighted average of its assigned features after Tasks 1 through 10. The results in Table 5 show that these similarity values remain in the range 0.600–0.850 and change smoothly as new classes are introduced. This indicates that the anchors stay close to the feature distributions.

**Anchor Stability.** We next examine how each anchor evolves over the task sequence. We compute the cosine similarity between the same anchor after consecutive tasks and average this value across all anchors. This diagnostic measures the temporal consistency of the anchors and is distinct from the anchor–feature similarity reported above. The values in Table 6 show that the anchors change smoothly.

Table 6: Anchor stability across tasks.

| Metric | Value |
|--------|-------|
| Mean stability | 0.823 |
| Max stability | 0.972 |
| Min stability | 0.642 |
| Anchors with stability $> 0.7$ | 79% |

This behaviour agrees with the exponential moving average update rule described in Section 3.1. These results further support the local coverage assumption that appears in Theorem 2.

**Semantic Coherence.** We also study the semantic coherence of the anchors. For each anchor, we retrieve the Tiny-ImageNet classes whose mean features are closest to the anchor. We then describe the shared visual patterns in these classes. The results in Table 7 show that the anchors remain aligned with coherent semantic groups throughout the entire training process. These groups include rigid objects, animals, background textures, and scenes with clear open-space patterns. This behaviour suggests that the anchor space organizes features in a stable and interpretable way as tasks accumulate. For classes marked with an asterisk, the descriptive terms refer to shared visual textures such as sky or water rather than official Tiny-ImageNet labels.

Across all diagnostics, the anchors remain close to routed features, evolve smoothly across tasks, and preserve meaningful semantic structure. These observations support the practical validity of the coverage assumption in Theorem 2. They also show that the anchor space maintains stable and interpretable behaviour throughout the full task sequence.

