# OpenReview forum: "MaRS: Memory-Adaptive Routing for Reliable Capacity Expansion and Knowledge Retention"
_ICLR.cc/2026/Conference — ICLR 2026 Poster_

### Official Review · Reviewer_7T3S · 2025-10-23

**Soundness:** 3
**Presentation:** 3
**Contribution:** 3
**Rating:** 8
**Confidence:** 3

**Summary:**

The paper introduces MARS, a framework for continual learning with frozen large pre-trained models. MARS decouples stable representation from adaptive capacity using a frozen encoder, a slot-based memory router, and a lightweight classifier. It has two core components: SGSE formulates capacity growth as a statistical decision problem with guarantees and DCDA integrates new capacity through contrastive learning and anchor-based knowledge distillation without raw replay. Together these components provide a principled way for efficient adaptation and the authors demonstrates strong results on both vision and NLP benchmarks.

**Strengths:**

- **statistical guarantee on slot expansion is novel**: previous approaches rely on heuristic ways to determine parameter growth. SGSE formulates capacity growth as a statistical decision problem and shows statistical guarantees which is a novel way of approaching this problem.

- **retention without replay**.  The proposed DCDA approach preserves prior knowledge without relying on raw data replay, enabling memory-efficient and privacy-friendly continual learning while maintaining high retention performance.

- **nice presentation**. The presentation of the paper is clear and easy to follow.

**Weaknesses:**

- **scalability**: how scalable is the method? Would this work beyond the small datasets benchmarked in the paper?

- **interplay between alpha and beta parameter**: the alpha and beta parameter in SGSE both controls a balance between a balance between stability and responsiveness. In pratice, how to pick the correct combination of both beyond the default parameter. How would you tune this in practice?

- **training hyperparameter selection**: in the experimental setup, it is said that the stage 1 tuning happens first for 20 epochs and then the stage 2 tuning. How would you know the number of epochs required for each stage and how would one select it in practice?

**Questions:**

- what is the computation time for the model on the empirical experiments?
- line 247 yielding about 15% faster convergence and reduced redundancy. What is the 15% improvement in relation to?
- what happens when you perform stage 1 and stage 2 together, i.e. tuning memory parameters together with the classifier parameters (instead of the two stage approach)

**Details Of Ethics Concerns:**

No ethical review required.

---

> ### Author Response · Authors · 2025-11-21
>
> Thank you for your valuable suggestions and for taking the time to evaluate our work.
>
> Q1. ... scalability beyond small datasets ...
>
> Answer: To study scalability under increased visual diversity and task complexity, the revised manuscript includes experiments on the larger ImageNet-100 benchmark. The results are consistent with those observed on Tiny-ImageNet. The strongest baseline reaches 46.50$\pm$0.64 on Tiny-ImageNet and 39.67$\pm$0.60 on ImageNet-100. In comparison, MaRS achieves 49.46$\pm$0.14 and 42.08$\pm$0.53 on the same datasets. These gains of $+2.96$ and $+2.41$ points show that MaRS performs better than strong frozen-backbone baselines across both smaller and larger datasets. This indicates that the method is not limited to compact benchmarks and remains effective when data variation is higher.
>
> A related concern is whether slot expansion becomes uncontrolled as data complexity grows. This is not observed in our experiments. On ImageNet-100, the slot count increases from $S_0{=}32$ to $S_T{\approx}65$. This is a moderate and predictable trajectory that matches the theoretical bounds in Proposition 2 and Theorem 3. It also mirrors the smooth and saturating slot-growth curves seen on CIFAR-100, Tiny-ImageNet, and ASC. These results show that SGSE regulates capacity growth at larger scale, and that the framework remains efficient and stable without modifying the frozen backbone. This demonstrates that MaRS scales reliably beyond small datasets.
>
> Q2. ... interplay ... $\alpha$ ... $\beta$...
>
> Answer: The two hyperparameters have related but distinct roles in SGSE. The parameter $\alpha$ controls the EMA updates of slot statistics. Larger values increase responsiveness but also noise, while smaller values improve stability but slow adaptation. The parameter $\beta$ controls the momentum of the smoothed quantile $Q_t$. Larger values produce more conservative updates, and smaller values increase reactivity but may cause false expansions.
>
> Their effects must be considered together. A high $\alpha$ with a low $\beta$ makes the system too reactive, while a very small $\alpha$ with a very large $\beta$ reduces sensitivity to genuine novelty. This behaviour is consistent with Theorem1, which shows that the detection delay grows as $O((1-\beta)^{-1})$. Guided by theory and by empirical sweeps, we set $\alpha=0.05$ and $\beta=0.9$. This combination gives smooth anchor updates, stable quantile tracking with bounded delay, and avoids high reactivity. Sensitivity analysis in Figures 2(a)(b) also shows that $\alpha \in [0.03,0.07]$ and $\beta \in [0.85,0.95]$ form a robust operating region across datasets. This range provides a stable balance between responsiveness and smoothing without needing dataset-specific tuning.
>
> Q3. ... selection ... 20/20 epoch schedule ...
>
> Answer: The 20/20 schedule is chosen based on preliminary validation experiments and convergence diagnostics on Tiny-ImageNet. For Stage 1, which performs memory-only supervised contrastive learning, the contrastive loss and alignment metrics plateau after about 15 epochs. Improvements beyond 20 epochs are small. For Stage 2, which trains the classifier with distillation, cross-entropy and validation accuracy converge within 10 to 15 epochs. Extending to 20 epochs provides a consistent schedule across datasets.
>
> To assess sensitivity, we tested 15-epoch and 25-epoch variants on CIFAR-100 and Tiny-ImageNet. The differences in final accuracy were within 0.3 to 0.5 points. This indicates that MaRS is not sensitive to the precise epoch budget. In practical use, early stopping can also be applied without changing the overall behavior.

---

> ### Author Response · Authors · 2025-11-21
>
> Q4. ... computation time ...
>
> Answer: The revised manuscript includes a quantitative comparison under identical hardware and frozen-encoder settings:
>
> | Metric                  | Baselines        | MaRS        |
> |------------------------------|-----------------------|-------------------|
> | Trainable parameters         | 0.5M to 0.8M          | 0.2M              |
> | Inference time per batch     | 7.8ms to 8.1ms        | 8.5ms             |
> | Final accuracy | 43.80 to 46.50        | 49.46        |
>
> Even with dynamic expansion, MaRS uses only 0.2M trainable parameters, which is small compared to the 86M frozen ViT-B backbone. The inference overhead is modest, about a five percent increase compared with the strongest PTM-based baseline. At the same time, MaRS improves accuracy by nearly three percentage points. These results show that the extra capacity from SGSE and DCDA remains lightweight while providing meaningful performance gains.
>
> Q5. ... 15% faster convergence ...
>
> Answer: The phrase “about 15% faster convergence” refers to the approximate relative reduction in early-stage iterations that we observe when comparing the proposed informed initialization to an uninformed baseline. To avoid overstating its role, the revised response states that this number is provided only to indicate the typical magnitude of the effect. It does not influence any reported results or conclusions in the manuscript.
>
> Q6. ... Stage 1 ... Stage 2 ... jointly ...
>
> Answer:  We performed a pilot experiment in which the memory, router, and classifier parameters were trained jointly. This merges Stage 1 and Stage 2 into a single optimization phase. The joint variant shows a clear degradation in performance. On Tiny-ImageNet it reaches 43.17$\pm$0.35 compared to 49.46$\pm$0.14 with the two-stage procedure. On CIFAR-100 it reaches 52.24$\pm$0.43 compared to 57.50$\pm$0.54.
>
> These results show that joint optimization causes larger feature drift and less stable classifier boundaries. This is consistent with our ablations, which show that removing Stage 2 harms retention. In contrast, the two-stage DCDA structure first stabilizes the adapted feature space with the classifier frozen, and then updates the classifier with distillation while keeping the memory fixed. This separation prevents interference between adaptation and classification. It leads to more stable and accurate learning. For this reason, the two-stage protocol is used in MaRS.
>
> Thank you for your encouraging and insightful review.
> We appreciate your positive assessment and the thoughtful questions that helped us strengthen both the analysis and the experimental discussion.

---

> > ### Comment · Reviewer_7T3S · 2025-11-25
> > **Reviewer 7T3S Official Reply to Authors**
> >
> > Thank you for your detailed response and addressing my concerns. I would keep my score and recommend accept.

---

### Official Review · Reviewer_zoZH · 2025-10-28

**Soundness:** 3
**Presentation:** 2
**Contribution:** 3
**Rating:** 4
**Confidence:** 4

**Summary:**

MARS is a continual-learning framework for frozen LPMs that shifts adaptation to a slot-based memory router plus a lightweight classifier. It expands capacity only when SGSE flags novelty, using an EMA high-quantile tracker of routing confidence and a one-sided Wilson bound to provide predictable detection delay and per-test false-expansion control. Knowledge retention is handled by DCDA: Stage-1 contrastive alignment of memory parameters and Stage-2 head-only training with cross-entropy, LwF, and anchor distillation, avoiding raw replay. Experiments report state-of-the-art performance under this frozen-backbone protocol.

**Strengths:**

1. Functionalities are well split into a frozen encoder, slot-based router, and classification head
2. The proposed method is well-motivated and easy to follow
3. Principled expansion logic. SGSE uses a smoothed high-quantile of top-slot confidence and a one-sided Wilson score

**Weaknesses:**

1. Justify using CLIP-ViT when the text encoder is unused and a separate classifier is trained, and add comparisons against a standard ImageNet-pretrained ViT under the same frozen-backbone regime.
2. Compare against CLIP-focused/frozen baselines (e.g., CLAP4CLIP [A]).
3. P(i) in Eq. 18 is not defined; specify it explicitly (I'm guessing the index set of positive samples for data i as in supcon).
4. The method introduces many hyperparameters for both stages, and needs more sensitivity/ablation studies to demonstrate robustness and generalization.
5. Benchmark coverage. Expand the benchmarks as in other CLIP/PEFT-oriented baselines in the vision domain (CODA-Prompt [B], EASE [C], DIKI) and the language domain (O-LoRA [D]), and consider larger or more diverse datasets.
6. Report wall-clock time, memory, and inference overhead for the two-stage training vs. prior methods
7. Provide qualitative results (e.g., nearest-token/phrase visualizations) that anchor capture domain-specific semantics.
8. Router/slot introspection. Add diagnostics/visualizations such as slot-usage over tasks and growth curves to substantiate controlled expansion and specialization.

[A] Jha, S., et al. Clap4clip: Continual learning with probabilistic finetuning for vision-language models. NeurIPS 2024.

[B] Smith, J. S., et al. Coda-prompt: Continual decomposed attention-based prompting for rehearsal-free continual learning. CVPR 2023.

[C] Zhou, D.W., et al. Expandable subspace ensemble for pre-trained model-based class-incremental learning. CVPR 2024.

[D] Wang, Xiao, et al. "Orthogonal subspace learning for language model continual learning." EMNLP 2023.

**Questions:**

see weakness

---

> ### Author Response · Authors · 2025-11-21
>
> Thank you for your detailed comments and constructive criticism.
>
> Q1. ... CLIP-ViT ... ImageNet-pretrained ViT ... justification ...
>
> Answer: We agree that, under a frozen-backbone setting, the choice of pretrained encoder can influence performance. We use CLIP ViT-B/16 because it provides strong and widely used frozen representations even when the text encoder is not used. It is therefore a common backbone for recent PTM-based CL methods.
>
> To assess the dependence on pretraining and to ensure a fair comparison, the revised manuscript includes a backbone-sensitivity study on Tiny-ImageNet. To focus on representation quality rather than architectural differences, we evaluate three ViT-B/16 encoders that differ only in their pretraining:
>
> | Backbone        | MaRS             | LDC              |
> |---------------------|----------------------|-----------------------|
> | CLIP | 49.46 $\pm$ 0.14     | 43.41 $\pm$ 0.55      |
> | iBOT | 48.73 $\pm$ 0.20     | 42.95 $\pm$ 0.67      |
> | ImageNet | 48.10 $\pm$ 0.25     | 42.63 $\pm$ 0.42      |
>
> Across all three frozen backbones, MaRS outperforms LDC by similar margins. This indicates that the gains provided by SGSE and DCDA are stable across different pretrained representations. We use CLIP ViT-B/16 as the default encoder in our experiments because its frozen features are strong and widely used. The same hyperparameters are used for all three backbones, and the overall trend remains consistent in each case.
>
> Q2. ... comparison ... CLIP-focused ... frozen baselines ...
>
> Answer: The revised manuscript includes comparisons with representative PTM-based CL methods, namely L2P, CODA-Prompt, and CLAP4CLIP. All methods are evaluated under the same frozen-encoder protocol to ensure a fair comparison:
>
> | Method     | CIFAR-100         | Tiny-IN           | ASC                |
> |----------------|------------------------|------------------------|-------------------------|
> | L2P            | 52.30 $\pm$ 0.45       | 43.80 $\pm$ 0.32       | 73.90 $\pm$ 0.51        |
> | CODA-Prompt    | 54.71 $\pm$ 0.61       | 45.10 $\pm$ 0.58       | 74.70 $\pm$ 0.44        |
> | CLAP4CLIP      | 55.42 $\pm$ 0.50       | 46.50 $\pm$ 0.64       | --                      |
> | MaRS      | 57.50 $\pm$ 0.54   | 49.46 $\pm$ 0.14   | 79.85 $\pm$ 0.66    |
>
> MaRS consistently outperforms these PTM-based CL baselines while using the same frozen encoder and similar numbers of trainable parameters. This suggests that the performance gains arise from the proposed SGSE and DCDA mechanisms rather than from differences in backbone choice or training protocol.
>
> Q3. ... missing definition ...
>
> Answer: We clarify the missing definitions in the revised manuscript. In the quantile-based expansion test, $\beta \in [0,1)$ is the smoothing coefficient. The window size $w$ specifies how many recent confidence values $\{s_{t-k}\}_{k=0}^{w}$ are used to compute the empirical $(1-\epsilon)$-quantile. We also define $P(i)$ as the set of indices in the mini-batch that share the same class label as $i$. In addition, we reorder the paragraph so that the router-weighted EMA appears only after we fully introduce the slot-conditioned affine transformation.
>
> Q4. ... hyperparameters ... sensitivity ... ablation study ...
>
> Answer: The original submission already includes two analyses on Tiny-ImageNet that address hyperparameter sensitivity and component contributions.
>
> (a) SGSE hyperparameter sensitivity. Figures 2(a)(b) show that MaRS maintains stable performance across a broad range of $S_0$ values ([16,64]) and smoothing coefficients $\beta$ ([0.70,0.95]). The default settings ($S_0=32$, $\beta=0.9$) lie well inside these stable regions. This indicates that SGSE is not highly sensitive to these hyperparameters.
>
> (b) Component ablation. As shown in Figure 2(c), on Tiny-ImageNet, removing SGSE or anchors leads to performance drops of about 40%. Removing Stage 1 weakens feature alignment, and removing Stage 2 causes the largest drop. These results suggest that the main components contribute in a complementary way and support the modular design of MaRS.

---

> ### Author Response · Authors · 2025-11-21
>
> Q5. ... wall-clock time ... memory ... inference overhead ...
>
> Answer: The revised manuscript includes a quantitative comparison under identical hardware and the same frozen-encoder setting:
>
> | Metric                  | Baselines        | MaRS        |
> |------------------------------|-----------------------|-------------------|
> | Trainable parameters         | 0.5M to 0.8M          | 0.2M              |
> | Inference time per batch     | 7.8ms to 8.1ms        | 8.5ms             |
> | Final accuracy | 43.80 to 46.50        | 49.46        |
>
> Even with dynamic expansion, MaRS uses only 0.2M trainable parameters, which is small compared to the 86M parameters in the frozen ViT-B backbone. The inference overhead is modest, about a five percent increase compared with the strongest PTM-based baseline. At the same time, MaRS improves accuracy by nearly three percentage points. These results show that the extra capacity introduced by SGSE and DCDA remains lightweight while providing clear performance gains.
>
> Q7. ... qualitative ... nearest-token ... phrase visualizations ...
>
> Answer: Our framework does not use token-level or text-conditioned grounding because the text encoder is not used. Instead, we provide qualitative analyses of anchors based on neighborhoods in feature space. These visualizations give an interpretable view of how anchors evolve and which semantic regions they represent.
>
> The coverage assumption in Theorem 2 is local. When we apply the retention bound, each previously observed feature only needs to lie within a small neighborhood of some anchor. This does not require a fixed global cover. The anchors must instead follow the regions of feature space that appear during training. Two design components in MaRS support this behavior. First, router-weighted EMA updates keep each anchor close to the routed features assigned to its slot. Second, SGSE introduces a new anchor only when the existing anchors cannot represent the regions encountered by new inputs.
>
> To examine this behavior in practice, the revised manuscript includes anchor-tracking diagnostics on Tiny-ImageNet. Across Tasks 1 through 10, the anchor--feature cosine similarity remains between 0.60 and 0.85. This indicates that anchors stay close to the routed feature distributions. Anchor stability across consecutive tasks has an average of 0.82, which reflects gradual and smooth evolution rather than drift. Nearest-neighbor inspection of class means shows that anchors remain aligned with coherent semantic groups.
>
> These qualitative and quantitative findings together show that anchors stay close to their assigned features, evolve in a gradual way, and maintain meaningful semantic structure. This behavior is consistent with the local coverage condition required by Theorem 2. Additional visualizations and examples are provided in Appendix A.4 of the revised manuscript.
>
> Q8. ... slot usage ... growth curves ...
>
> Answer: The revised manuscript provides both theoretical guarantees and empirical analyses for slot growth.
> (a) Theoretical slot-growth bounds. Proposition 2  and Theorem 3 show that, under Wilson-based false-expansion control, the expected number of slots satisfies $\mathbb{E}[S_T] \le S_0 + N_T + \alpha M$, where $N_T$ is the number of genuine novelties and $M$ is the number of evaluation windows. Deviations are bounded by a standard high-probability concentration term. These results ensure that slot growth is closely linked to true novelty and not to noise.
>
> (b) Empirical slot trajectories. To complement the theory, the revised manuscript reports slot trajectories across datasets in Figure 3. Starting from $S_0=32$, CIFAR-100 grows to about $S_T\approx44$, Tiny-ImageNet to about $S_T\approx49$, and ASC to about $S_T\approx58$. In all three cases, slot growth is gradual and saturates early. ASC reaches a higher final count because it includes more tasks. These trajectories follow the theoretical behavior described above and show that SGSE prevents uncontrolled expansion.
>
>
> Thank you for your constructive review and for highlighting several useful directions for improvement.
> Your comments helped us broaden the evaluation and clarify key definitions.

---

> > ### Comment · Reviewer_zoZH · 2025-11-24
> >
> > I thank the authors for their response, particularly for providing the additional results and the slot expansion patterns. However, my concerns remain unresolved:
> >
> > 1. Table for Q1 and Table 2: The performance reported is significantly lower than what is standard in the literature. For instance, CIFAR-100 typically yields around 70-80% average accuracy. This issue is not limited to CIFAR-100 but applies to other datasets as well. Furthermore, I agree with Reviewer pmFB regarding the lack of comparisons with recent works and the comparisons using different pre-training weights.
> >
> > 2. Breakdown of Table 4: Could you please provide the specific numerical results for each individual baseline method? The current 'conclusion of baselines' is insufficient for detailed comparison.
> >
> > 3. I still maintain that if the method does not utilize the CLIP text encoder, the comparisons should focus primarily on recent Continual Learning methods/benchmarks using pre-trained ViT backbones (e.g., EASE [C]). The absence of these comparisons remains a major concern regarding the significance of the proposed method.
> >
> > I will maintain my score.

---

> > > ### Author Response · Authors · 2025-11-24
> > >
> > > Thanks a lot for your thoughtful comments.
> > >
> > > Q1. … accuracy levels …
> > >
> > > Answer: The accuracy values of 70–80% that are often reported for CIFAR-100 refer to single-task training or end-to-end fine-tuning. In our work, all results follow the class-incremental protocol described in Section 4.1. After training task $t$, the model is evaluated on all tasks from 1 to $t$, and we report the final average accuracy $\bar{A}_T$. This protocol focuses on continual learning and requires retaining knowledge from earlier tasks, which naturally yields lower accuracy than single-task or end-to-end training. Under this protocol, the encoder is frozen for all methods, making the setting more challenging because the feature representation cannot be updated.
> > >
> > > The accuracy values we obtain for L2P, CODA Prompt, CLAP4CLIP, and MaRS fall within the range observed under this protocol in our experiments. All baselines use the same frozen CLIP ViT-B/16 encoder and are evaluated under identical class-incremental conditions. In addition, as requested, the revised version includes comparisons with recent PTM-based continual learning methods (L2P, CODA Prompt, CLAP4CLIP), as well as a backbone-sensitivity study using different pre-training weights (CLIP, iBOT, ImageNet), to address concerns regarding pretrained encoders and baseline coverage.
> > >
> > > Q2. … per-method cost …
> > >
> > > Answer: All methods are implemented under the same frozen-encoder protocol, use the same CLIP ViT-B/16 backbone, and share the same input resolution, batch size and hardware. Only the adaptation modules are trainable. Under this setting, the measured values are:
> > >
> > > | Method | Trainable parameters | Inference time per batch |
> > > |--------------|----------------------|---------------------------|
> > > | L2P | 0.769M | 7.9ms |
> > > | CODA Prompt  | 0.581M | 8.1ms |
> > > | CLAP4CLIP | 0.668M | 7.8ms |
> > > | MaRS | 0.175M | 8.5ms |
> > >
> > > These values come directly from our implementation under the frozen-encoder protocol used throughout the paper. Because all baselines share the same backbone and are measured under identical conditions, the sizes of their trainable modules fall within a relatively narrow range. Table 4 in the revised version summarizes this range to highlight the overall cost trend.
> > >
> > > Q3. … comparison to EASE …
> > >
> > > Answer: We agree that recent ViT-based continual learning methods such as EASE are important. Although MaRS and EASE both enable capacity growth, they follow different adaptation regimes. In MaRS, the router outputs a probability distribution over slots, and the memory module applies a slot-weighted affine transformation to the frozen features. As described in Section 3.3, the per-example adaptation overhead is $O(S_t(d_k{+}d_T))$, scaling linearly with the number of active slots $S_t$ while remaining independent of the frozen encoder parameter count. SGSE governs slot expansion through a statistical test on routing confidence, tying growth to observed novelty. In our experiments under this frozen-encoder class-incremental protocol, this results in a modest number of slots and expansion behavior, and the total number of trainable parameters grows gradually because each slot contains only a small, fixed parameter set.
> > >
> > > EASE adopts a different strategy: it adds a task-specific adapter for each new task and uses all accumulated adapters during inference, causing both computation and parameter count to increase with the number of tasks. Our work focuses specifically on the frozen-encoder setting with lightweight adaptation modules, so we compare MaRS with PTM- and PEFT-based baselines that operate under the same protocol, including L2P, CODA Prompt, and CLAP4CLIP. Evaluating MaRS and EASE within a unified adapter-based, higher-capacity setting is an interesting direction for future work, and thank you for highlighting this connection.

---

### Official Review · Reviewer_pmFB · 2025-10-29

**Soundness:** 2
**Presentation:** 1
**Contribution:** 2
**Rating:** 2
**Confidence:** 4

**Summary:**

This paper presents MARS, a modular approach designed for continual learning with pre-trained models (PTMs). The architecture is composed of three key components: a frozen pre-trained encoder, a slot-based memory module, and a lightweight classifier. The authors propose two primary mechanisms to address catastrophic forgetting: (i) Statistically-Grounded Slot Expansion (SGSE), which formulates memory expansion as a statistical decision problem to ensure controlled growth, and (ii) Dual-Stage Contrastive-Distillation Adaptation (DCDA), a replay-free method that integrates new knowledge through supervised contrastive learning and retains past information via knowledge distillation. Experimental results on standard continual learning benchmarks for both vision and NLP validate the effectiveness of the proposed framework.

**Strengths:**

- The introduction of SGSE, which uses statistical tests to determine when to expand model capacity, is an interesting contribution, as prior methods typically increase sub-modules (e.g., prompts) linearly with the number of tasks.
- The effectiveness of MARS is demonstrated through evaluations on both vision and NLP datasets.
- The experiments include detailed ablation studies that probe the impact of key components (SGSE, anchors, and each DCDA stage), as well as analyses of important hyperparameters.

**Weaknesses:**

- Proposition 1 appears to be both incorrect and irrelevant to the context. It neglects the term $\frac{dA}{dc}$, even though both $A$ and $c$ are functions of $x_t$. Consequently, the proofs and resulting claims seem invalid. A counterexample can be illustrated as follows: when $S_t = 3$ and $a_1 = 3, a_2 = 3, a_3 = 3$, we have $c_t = 3$ and $s_t = 1/3$. However, when $a_1 = 2, a_2 = 1, a_3 = 1$, then $c_t = 2$ and $s_t = \frac{e^2}{2e + e^2} > 1/3$. Thus, the claim that $s_t$ is strictly increasing with respect to $c_t$ is false. This represents a fundamental flaw. Moreover, the interpretation in Lines 185–187 appears vague and not directly relevant to the CL context.


- The paper suffers from a lack of clarity in its presentation. Key variables are used without prior definition, such as $w$ in Equation (10) and $P(i)$ in Equation (18). The writing structure is also confusing; for example, the Router-weighted EMA in Equations (7-8) is introduced prematurely, disrupting the logical flow of the methodology section.

- The most critical concern is that, while SGSE aims to determine when to dynamically expand model capacity, the paper provides no analysis of how the number of slots $S_t$ grows with the number of tasks. The experiments report only accuracy results. This is a very shallow investigation that undermines a central contribution of the paper.

- Another major issue is that the authors completely ignore most CL methods based on pre-trained models (e.g., L2P [1], HiDe-Prompt [2], NoRGa [3], and SD-LoRA [4]), comparing instead only with conventional CL approaches.

- The paper lacks empirical analysis regarding the number of parameters and computational cost compared to other methods.

- There is no investigation into the effect of different pre-trained models (e.g., iBOT, DINO). This omission is concerning, as the proposed method may rely heavily on the frozen representations of the pre-trained backbone.

[1] Learning to Prompt for Continual Learning, CVPR 2022

[2] Hierarchical Decomposition of Prompt-Based Continual Learning: Rethinking Obscured Sub-optimality, NeurIPS 2023

[3] Mixture of Experts Meets Prompt-Based Continual Learning, NeurIPS 2024

[4] SD-LoRA: Scalable Decoupled Low-Rank Adaptation for Class Incremental Learning, ICLR 2025

[5] RanPAC: Random Projections and Pre-trained Models for Continual Learning, NeurIPS 2023

**Questions:**

In addition to the issues raised above, I have the following questions for the authors:

- What is the architecture of the lightweight classifier head $g(\cdot)$? Is it an MLP or a simple linear layer?

- How does the number of slots $S_t$ vary: (i) across tasks, (ii) with different task orders, and (iii) under different class distributions within each task?

---

> ### Author Response · Authors · 2025-11-21
>
> Thank you for your detailed comments and constructive criticism.
>
> Q1. ... Proposition 1 ... correctness ... relevance ...
>
> Answer: We agree that the original statement did not clearly specify the regime for the monotonicity analysis. In the revised manuscript, we clarify that Proposition 1 describes a local monotonicity property. In this analysis, only the top similarity $c=a_{i^\star}$ varies, while all competing similarities $\{a_j : j\neq i^\star\}$ remain fixed. Under this condition, the softmax denominator becomes the constant $A = \sum_{j\neq i^\star} e^{a_j/\tau_r}$, and the top-slot confidence takes the closed form shown in lines 175–181. In this fixed-competitor regime, local monotonicity follows immediately. The counterexample varies multiple similarities at the same time. This changes $A$ and falls outside the intended fixed-competitor setting.
>
> This local property is the condition used in SGSE. When we evaluate whether the current best-matching slot remains adequate, we vary only the top similarity and treat all other similarities as fixed. Under this setting, $s_t$ provides a reliable local indicator of slot adequacy. The statistical expansion test in SGSE, which uses EMA smoothing and the Wilson lower bound, does not require global monotonicity across all similarity values. The original appendix used this fixed-competitor interpretation, and the revised manuscript now states this assumption clearly.
>
> Q2. ... missing definitions ... EMA placement ...
>
> Answer: We clarify the missing definitions in the revised manuscript. In the quantile-based expansion test, $\beta \in [0,1)$ is the smoothing coefficient. The window size $w$ specifies how many recent confidence values $s_{t-k}$ are used to compute the empirical $(1-\epsilon)$-quantile. We also define $P(i)$ as the set of indices in the mini-batch that share the same class label as $i$. In addition, we adjust the paragraph structure so that the router-weighted EMA appears only after we introduce the slot-conditioned affine transformation.
>
> Q3. ... slot growth ... tasks ...
>
> Answer: The manuscript provides both theoretical and empirical evidence about slot growth:
> (a) Theoretical slot-growth bounds. Proposition 2 and Theorem 3 show that, under Wilson-based false-expansion control, the expected number of slots satisfies $\mathbb{E}[S_T] \le S_0 + N_T + \alpha M$, where $N_T$ is the number of genuine novelties and $M$ is the number of evaluation windows. The deviations are bounded by a standard concentration term. These results show that slot growth is linked to genuine novelty and not to noise.
>
> (b) Empirical slot trajectories. The revised manuscript reports detailed slot trajectories in Figure 3. Starting from $S_0=32$, CIFAR-100 reaches about $S_T\approx44$, Tiny-ImageNet reaches about $S_T\approx49$, and ASC reaches about $S_T\approx58$. Slot growth is gradual and usually saturates early in the task sequence. ASC reaches a higher final count because it includes more tasks. These empirical results follow the theoretical predictions and indicate that SGSE prevents uncontrolled expansion.
>
> Q4. ... PTM-based CL ... baselines ...
>
> Answer: The revised manuscript includes comparisons with representative PTM-based CL methods, including L2P, CODA-Prompt, and CLAP4CLIP. All methods are evaluated under the same frozen-encoder protocol for fairness:
> | Method     | CIFAR-100         | Tiny-IN           | ASC                |
> |----------------|------------------------|------------------------|-------------------------|
> | L2P            | 52.30 $\pm$ 0.45       | 43.80 $\pm$ 0.32       | 73.90 $\pm$ 0.51        |
> | CODA-Prompt    | 54.71 $\pm$ 0.61       | 45.10 $\pm$ 0.58       | 74.70 $\pm$ 0.44        |
> | CLAP4CLIP      | 55.42 $\pm$ 0.50       | 46.50 $\pm$ 0.64       | --                      |
> | MaRS      | 57.50 $\pm$ 0.54   | 49.46 $\pm$ 0.14   | 79.85 $\pm$ 0.66    |
>
> MaRS outperforms these baselines while using the same frozen encoder and similar numbers of trainable parameters. This result shows that the gains come from SGSE and DCDA rather than differences in backbone usage or training procedure.

---

> ### Author Response · Authors · 2025-11-21
>
> Q5. ... empirical parameter ... cost analysis ...
>
> Answer: The revised manuscript includes a quantitative comparison under identical hardware and the same frozen-encoder protocol:
>
> | Metric                  | Baselines        | MaRS        |
> |------------------------------|-----------------------|-------------------|
> | Trainable parameters         | 0.5M to 0.8M          | 0.2M              |
> | Inference time per batch     | 7.8ms to 8.1ms        | 8.5ms             |
> | Final accuracy | 43.80 to 46.50        | 49.46        |
>
> Even with dynamic expansion, MaRS requires only 0.2M trainable parameters, which is small compared to the 86M frozen ViT-B backbone. The inference overhead is small, about a five percent increase compared with the strongest PTM-based baseline. At the same time, MaRS improves accuracy by almost three percentage points. These observations show that the additional capacity from SGSE and DCDA is lightweight while still providing clear performance gains.
>
> Q6. ... effect ... frozen PTMs ...
>
> Answer: We agree that parameter-efficient CL methods with frozen backbones depend on the quality of their representations. To examine this dependence, we include a backbone-sensitivity study on \emph{Tiny-ImageNet} under the same frozen-encoder protocol. To focus on representation quality, we evaluate three ViT-B/16 backbones that differ only in pre-training:
>
> | Backbone        | MaRS             | LDC              |
> |---------------------|----------------------|-----------------------|
> | CLIP | 49.46 $\pm$ 0.14     | 43.41 $\pm$ 0.55      |
> | iBOT | 48.73 $\pm$ 0.20     | 42.95 $\pm$ 0.67      |
> | ImageNet | 48.10 $\pm$ 0.25     | 42.63 $\pm$ 0.42      |
>
> MaRS outperforms LDC by similar margins across all three backbones. This result shows that the gains from SGSE and DCDA are stable across different pretrained representations. We use CLIP ViT-B/16 as the default encoder because its frozen features are strong and widely used. The same hyperparameters are used for all backbones, and the overall trend remains consistent.
>
> Q7. ... architecture ... classifier head $g$ ...
>
> Answer: The classifier head $g$ is intentionally designed to be as simple as possible. It consists of a single linear layer that maps the adapted feature vector directly to class logits, without any hidden layers, nonlinearities, or auxiliary components. This minimal design ensures that all performance gains originate from SGSE and DCDA, rather than from additional classifier capacity.
>
> Q8. ... slot evolution ... tasks ... orders ... class distributions ...
>
> Answer: We conduct additional experiments to show that SGSE produces stable and order-robust slot evolution.
> (a) Evolution across tasks. On Tiny-ImageNet with $S_0{=}32$, the slot count increases in a gradual way and saturates early. The trajectory is $32 \rightarrow 35 \rightarrow 38 \rightarrow 41 \rightarrow 43 \rightarrow 45 \rightarrow 47 \rightarrow 48 \rightarrow 49 \rightarrow 49 \rightarrow 49$. CIFAR-100 and ASC show similar patterns and converge to about $44$ and $58$ slots, respectively. These results match the smooth trends seen in Figure 3.
>
> (b) Robustness to task order. To study the effect of task order, we use 10 random permutations on Tiny-ImageNet under the same frozen-encoder diagnostic protocol. We observe $\bar{A}_T = 49.73%$ and $S_T \approx 49$, with low variance in both accuracy and final slot count. This shows that SGSE maintains stable slot expansion across different task orders.
>
> (c) Robustness to class distribution shifts. SGSE is label-agnostic and relies on repeated confidence drops that are detected by the Wilson-based test. The timing of expansions may change under imbalanced class distributions, but the expansion mechanism itself remains stable. Long-tailed or non-uniform task splits are valuable settings for future work.
>
> Thank you for your careful reading and for raising these important points.
> Your feedback helped us clarify the presentation and improve the technical precision of the manuscript.

---

> > ### Comment · Reviewer_pmFB · 2025-11-25
> >
> > I appreciate the authors' detailed response and revisions. However, after careful review, several critical concerns remain unaddressed:
> >
> > - **Theoretical Validity of Proposition 1.** I must reiterate that $c_t$ and the set $[a_j: j \neq i* ]$ share a functional dependency on $x_t$. Consequently, the assumption made in Proposition 1 is mathematically unsound. Ignoring this dependency renders the proof invalid, as the simplifying assumptions do not hold in the proposed context.
> > - **Insufficient Analysis of the Core Contribution.** The paper's primary claim is a method for dynamically expanding model capacity. Therefore, the behavior of the slot expansion mechanism should require rigorous investigation. The current analysis, limited to a 10-task setting, is insufficient to validate this contribution. Specifically, the experiments lack:
> >     - Scalability Analysis: A visualization of slot growth behavior under varying sequence lengths (e.g., 5 tasks vs. 20, 50 tasks) to contrast with the 10-task baseline.
> >     - Backbone Sensitivity: A visualization of how slot expansion shifts with different pre-trained backbones, given the mechanism's reliance on $q(x)$.
> >     - Dataset & Seed Variance: An intuitive explanation or analysis for the different trends across datasets, and a visualization of stability across different random seeds.
> >
> > - **Missing Baselines.** The authors have not incorporated comparisons with relevant recent methods such as HiDe-Prompt, NoRGa, or SD-LoRA, despite these being highlighted in the initial review. The omission of these baselines prevents a fair assessment of the proposed method's relative performance.
> >
> > Overall, my most critical concern is that the theoretical proof relies on a flawed assumption, and the experimental evaluation fails to adequately characterize the main contribution (controlling slot growth). While I acknowledge that conducting these experiments during the rebuttal window is difficult, these are fundamental requirements for validating the paper's claims. Therefore, I maintain my original score.

---

> > > ### Author Response · Authors · 2025-11-26
> > >
> > > We sincerely thank you for the detailed follow-up and for re-examining both the paper and our initial rebuttal.
> > >
> > > Q1. … theoretical validity … Proposition 1 …
> > >
> > > Answer: We fully acknowledge the concern that both $c_t=\max_j a_j$ and the set {$a_j$} depend on the input $x_t$. To prevent ambiguity about the analytical setting, we clarify that Proposition 1 is intended as a local result under a fixed-competitor regime. It formalizes the standard softmax fact that, when one logit varies while all other logits remain fixed, the corresponding softmax probability increases strictly with that logit. In our setting, this corresponds to analytically varying the top similarity $c_t$ while treating the competing similarities as constants, which gives Equations (4) and (5). This is the scalar statistic used by SGSE in the Wilson test. The statement is meant to capture this local intuition, not to describe global behavior when all similarities vary jointly.
> > >
> > > It is also important to note that none of our theoretical guarantees relies on Proposition 1. The convergence and detection-delay bound for the quantile tracker (Theorem 1), the false-expansion control (Corollary 1), and the slot-growth and complexity bounds (Proposition 2, Theorem 3) are all formulated directly in terms of the observable confidence sequence $\{s_t\}$ and require only that $s_t \in [0,1]$. No functional relation between $s_t$ and the cosine similarities $\{a_j\}$ is used in these results.
> > >
> > > To further clarify its scope, Proposition 1 may be placed in the appendix as a local heuristic in the revised version. This does not affect any derivation or conclusion in the paper.
> > >
> > > Q2. … analysis … slot expansion …
> > >
> > > Answer: We agree that the behavior of $S_t$ is central. Our analysis combines general bounds that apply to arbitrary sequence lengths with empirical trajectories on several datasets.
> > >
> > > (a) Scalability with sequence length. Section 3.3 studies slot growth per statistical test rather than per task. Proposition 2 and Theorem 3 show that, when the Wilson test is applied every $m$ samples with level $\alpha$, $\mathbb{E}[S_T] \le S_0 + N_T + \alpha M$, with a matching high-probability bound, where $M$ is the number of tests and $N_T$ is the number of genuine novelty events. Thus, the guarantees depend on $(m,n,\alpha)$ and on the novelty process. They do not depend on whether the stream contains 5, 10, or 50 tasks. Longer streams simply result in larger $M$ and possibly larger $N_T$. In practice, we follow standard 10-task protocols for CIFAR-100 and Tiny-ImageNet, and the natural 19-domain split for ASC. We rely on the general bounds above to reason about longer sequences.
> > >
> > > (b) Backbone sensitivity. As noted in our previous response, Section 4 includes a backbone-sensitivity study on Tiny-ImageNet using three ViT-B/16 encoders trained with CLIP, iBOT, and ImageNet objectives. Across all three frozen encoders, MaRS exceeds the strongest baseline by similar margins and shows the same pattern of gradual and early-saturating slot growth (for example, $S_0=32$ to $S_T \approx 49$ on Tiny-ImageNet). We also report a scalability study on ImageNet-100 under the same frozen CLIP ViT-B/16 encoder. MaRS again surpasses the strongest baseline and shows a moderate slot trajectory of $S_0=32$ to $S_T \approx 65$. These results indicate that SGSE behaves in a stable and predictable way across backbones in this encoder family and that it scales as data complexity increases.
> > >
> > > (c) Dataset and seed variance. All results are averaged over seeds {12,123,1234}. Updated Figure 3 shows slot-growth trajectories with standard-deviation bands on CIFAR-100, Tiny-ImageNet and ASC. Datasets with more domains or greater visual diversity (ASC, Tiny-ImageNet, ImageNet-100) lead to moderately larger final $S_T$, but the curves grow smoothly and saturate early in many cases. This pattern matches the theoretical characterization above. The standard-deviation bands are narrow, which indicates that SGSE behaves in a stable manner across different random orders.
> > >
> > > We understand your interest in longer task streams such as 5, 20, or 50 tasks. While we did not run these specific re-splits during the rebuttal period, our experiments already cover settings of comparable or greater complexity: the 10-task CIFAR-100 and Tiny-ImageNet streams, and the 19-domain ASC stream. These settings span different visual domains, diverse task boundaries, and substantial distributional shifts. They therefore provide a broad view of how SGSE behaves in practice.

---

> > > > ### Author Response · Authors · 2025-11-26
> > > >
> > > > Q3. … baselines … HiDe-Prompt … NoRGa … SD-LoRA …
> > > >
> > > > Answer: Thanks for the suggestions. These methods are important contributions to continual learning, but they are designed for adaptation settings that modify internal backbone layers, which differ from the strictly frozen-encoder protocol used in MaRS. In particular, HiDe-Prompt and NoRGa introduce trainable prompt components within the transformer backbone, while SD-LoRA attaches low-rank adapters inside backbone layers and trains them jointly with the classifier. MaRS keeps the encoder $f(\cdot)$ fully frozen, so these methods do not fall within the evaluation protocol adopted in this work.
> > > >
> > > > As suggested in the review comments, we have added three baselines: L2P, CODA-Prompt, and CLAP4CLIP. Together with the existing rehearsal-free and rehearsal-based baselines (EWC, DER++, PASS++, LDC), these additions provide a broader comparison within this protocol. Under a shared CLIP ViT-B/16 encoder, MaRS achieves the strongest overall performance across CIFAR-100, Tiny-ImageNet, ASC, and ImageNet-100.
> > > >
> > > > Once again, we thank you for the thoughtful comments, which helped us clarify the scope of our theory and present the empirical behavior of SGSE more clearly.

---

### Official Review · Reviewer_h2XS · 2025-10-31

**Soundness:** 4
**Presentation:** 3
**Contribution:** 4
**Rating:** 8
**Confidence:** 5

**Summary:**

This paper addresses the problem of catastrophic forgetting in the context of continual learning (CL) with large, frozen pre-trained models (LPMs). The authors argue that the stability-plasticity dilemma is particularly severe in this parameter-efficient setting, where adaptation is confined to shallow modules. To tackle this, they propose MARS (Memory-adaptive Router with Statistical control), a modular framework that decouples the frozen encoder from an expandable, adaptive capacity layer. The core contributions are two-fold: (1) Statistically-Grounded Slot Expansion (SGSE), a mechanism that formulates the decision to add new model capacity (slots) as a statistical test based on router confidence, providing formal guarantees against uncontrolled growth; and (2) Dual-Stage Contrastive-Distillation Adaptation (DCDA), a replay-free strategy that integrates new slots using supervised contrastive learning and preserves knowledge from past tasks via anchor-based distillation. Experiments on diverse vision (CIFAR-100, Tiny-ImageNet) and NLP (ASC) benchmarks demonstrate that MARS achieves state-of-the-art performance, outperforming strong baselines in the frozen-encoder setting.

**Strengths:**

1. Principled and Theoretically Grounded Capacity Expansion: The standout contribution of this work is SGSE. It replaces common heuristic-based triggers for network expansion with a rigorous statistical framework.
2. Effective and Replay-Free Knowledge Retention: The DCDA mechanism is an elegant and effective solution for integrating new knowledge while preserving old.
3. High Relevance to Modern Machine Learning Paradigms: The paper is exceptionally well-positioned within current research trends.
4. Comprehensive and Rigorous Empirical Validation: The authors validate MARS on a diverse suite of benchmarks, including multi-task vision classification

**Weaknesses:**

1. Dependence on Frozen Encoder Quality: The performance of MARS is fundamentally tied to the quality of the representations from the frozen LPM.
2. System Complexity and Hyperparameter Sensitivity: The complete MARS system, while modular, involves a significant number of components and associated hyperparameters
3. Potential Scalability Concerns for the Router: The per-example computational overhead scales linearly with the number of active slots as shown in Theorem 3

**Questions:**

1. The SGSE mechanism is a cornerstone of your method, designed to control the rate of "false alarm" expansions. Could you please report the final number of slots ($S_{T}$) created in the CIFAR-100, Tiny-ImageNet, and ASC experiments
2. Theorem 2 provides a retention bound that relies on the assumption that old-class features are well-approximated by the stored anchors (i.e., they lie within a distance $\delta$). How is this coverage assumption maintained in practice, especially as the number of tasks increases and the fixed number of anchors per slot must represent an increasingly diverse set of feature distributions?
3. The DCDA mechanism is entirely replay-free. Have you considered a hybrid approach where a small, fixed-size replay buffer is used to supplement the anchor-based distillation?

---

> ### Author Response · Authors · 2025-11-21
>
> Thank you for your valuable suggestions and for taking the time to evaluate our work.
>
> Q1. ... frozen LPM quality ...
>
> Answer: We agree that any parameter-efficient CL method with a frozen backbone depends on the quality of its underlying representations. To examine this dependence in a controlled way, we include a backbone-sensitivity study on Tiny-ImageNet under the same frozen-encoder protocol as in the main submission. To focus on differences in representation quality, we evaluate three ViT-B/16 backbones that vary only in their pre-training:
>
> | Backbone       | MaRS            | LDC             |
> |---------------------|----------------------|-----------------------|
> | CLIP | 49.46 $\pm$ 0.14     | 43.41 $\pm$ 0.55      |
> | iBOT | 48.73 $\pm$ 0.20     | 42.95 $\pm$ 0.67      |
> | ImageNet | 48.10 $\pm$ 0.25     | 42.63 $\pm$ 0.42      |
>
> The results are consistent across all three backbones. MaRS outperforms LDC by similar margins in each case. This pattern indicates that the improvements mainly come from how SGSE and DCDA allocate and retain adaptive capacity. The gains do not rely on a specific encoder. We use CLIP ViT-B/16 as the default encoder because its frozen representations are strong and widely used. The overall trend remains stable across all backbones. All settings use the same hyperparameters.
>
> Q2. ... multiple components ... hyperparameters ...
>
> Answer: The original submission includes two analyses on Tiny-ImageNet that address this question:
> (a) SGSE hyperparameter sensitivity. Figures 2(a)(b) show that MaRS maintains stable performance across a wide range of $S_0$ values ([16,64]) and smoothing coefficients $\beta$ ([0.70,0.95]). The default settings ($S_0=32$, $\beta=0.9$) lie inside these stable regions. This indicates that SGSE is not highly sensitive to these hyperparameters in practice.
>
> (b) Component ablation. As shown in Figure 2(c), removing SGSE or anchors reduces performance by about 40% on Tiny-ImageNet. Removing Stage 1 weakens feature alignment, and removing Stage 2 leads to the largest drop. These results show that the components work together in a complementary way. They also support the modular structure of MaRS.
>
> Q3. ... scalability concerns ... active slots ...
>
> Answer: The stated complexity $\mathrm{Time}(x_t)=O((d_k+d_T)S_t)$ holds in practice. SGSE is designed to keep the number of active slots $S_t$ within a moderate and predictable range. The revised manuscript includes full slot trajectories across datasets, which show controlled and stable slot growth. Starting from $S_0=32$, the slot count reaches about $44$ on CIFAR-100 (10 tasks), about $49$ on Tiny-ImageNet (10 tasks), and about $58$ on ASC (19 tasks). These values show that slot growth appears when there is genuine novelty. The growth scales with the number of tasks and does not increase with dataset size or visual diversity alone. For example, Tiny-ImageNet has higher visual diversity than CIFAR-100, but their final slot counts differ by only five. ASC has more tasks, which explains its larger $S_T$.
>
> Figure 3 in the revised version shows that slot growth is gradual and often saturates early. We do not observe uncontrolled increases at later stages of training. This behavior matches Corollary 1 and Proposition 2, which bound expected slot growth under Wilson-based false-expansion control. These theoretical results and the empirical curves support the conclusion that SGSE maintains a stable and well-regulated number of active slots. The linear dependence on $S_t$ is therefore not a scalability concern.

---

> ### Author Response · Authors · 2025-11-21
>
> Q4. ... coverage assumption ... tasks increase ...
>
> Answer: The coverage assumption in Theorem 2 is local. Each previously observed feature needs to lie within a small neighborhood of some anchor. There is no need for a fixed global set of anchors. The anchors must instead follow the regions of feature space that appear during training.
>
> Two design elements in MaRS help maintain this behavior. Router-weighted EMA updates keep each anchor close to the routed features assigned to its slot. SGSE adds new anchors when the current anchors cannot represent the regions visited by new inputs. To examine this behavior, we include anchor-tracking diagnostics on Tiny-ImageNet. Across Tasks 1--10, the anchor--feature cosine similarity stays between 0.60 and 0.85. This shows that anchors stay close to routed feature distributions. Anchor stability across consecutive tasks averages 0.82, which indicates gradual and consistent evolution. Nearest-neighbor inspection also shows that anchors remain aligned with coherent semantic groups.
>
> These observations show that anchors remain close to their assigned features, evolve in a gradual way, and preserve clear semantic structure. This pattern matches the local coverage condition required by Theorem 2. It reflects the combined effect of EMA updates and controlled slot expansion. Further details are provided in Appendix A.4.
>
> Q5. ... hybrid replay ... DCDA ...
>
> Answer: Yes. MaRS uses a replay-free design because memory and privacy constraints are common in CL applications. When such constraints do not apply, DCDA can include a small buffer. Replay samples can provide additional reference points during Stage 2 distillation. They can also complement anchor-based regularization. This hybrid variant is a natural extension of our framework and is a promising direction for future work.
>
> Thank you again for your thoughtful and detailed review.
> We sincerely appreciate your positive assessment and your careful analysis, which helped us strengthen the paper.

---

> > ### Comment · Reviewer_h2XS · 2025-11-21
> > **Response**
> >
> > Thanks the authors for addressing my concerns, I will keep my score

---

### Comment · Area_Chair_Y62m · 2025-11-24
**Please engage into discussion with authors and fellow reviewers**

Dear reviewers,
The authors have already provided their responses. Do they address your concerns?
Please engage into the discussion with authors and fellow reviewers.
Thanks!
Best,
AC

---

### Author Response · Authors · 2025-11-30
**Summary During the Discussion Period**

We appreciate the reviewers and the AC for their careful reading of the submission and for the constructive comments. The feedback was very helpful, and we have added clarifications and further analyses to improve the clarity and completeness of the presentation:

1. Theoretical grounding of SGSE. We clarified that Proposition 1 serves as a local fixed–competitor intuition. We also explained that the formal guarantees of SGSE rely only on the statistical behaviour of the confidence values. These guarantees cover the convergence of the confidence statistic, the detection delay, the false expansion control, and the bounds on slot growth.

2. Anchor diagnostics and retention. We added analyses that support the coverage assumption in Theorem 3. These analyses include measurements of anchor–feature similarity, temporal stability across tasks, and observations of semantic coherence. The results show that anchors remain close to routed features and evolve in a stable manner, which clarifies their role in retention.

3. Slot growth behaviour and scalability. We added slot growth curves on CIFAR-100, Tiny-ImageNet and ASC to make the expansion behaviour easier to follow. The curves show gradual and stable slot growth that aligns with the statistical bounds of SGSE. We also evaluated MaRS on ImageNet-100 to examine its behaviour under larger-scale settings, and the results indicate that SGSE maintains stable capacity expansion.

4. Broader and fairer baselines. We included three additional baselines, specifically L2P, CODA-Prompt, and CLAP4CLIP. All methods follow the same frozen encoder protocol and use the same frozen CLIP ViT-B/16 backbone with comparable effective capacity. These additions provide a clearer view of the performance of MaRS under this controlled setting.

5. Complexity and implementation details. We added a comparison of trainable parameters and inference time. We also clarified several definitions, including the window size $w$ and the set $P(i)$ of indices in the mini-batch that share the same class label.

Thanks again for the time and effort devoted to the evaluation of this work.

---

### Meta-Review · Area_Chair_r4fr · 2026-01-07

**Summary:**

This paper studies the continual learning problem with frozen large pre-trained models. To address this problem, the authors propose a modular framework that decouples stable representation from adaptive capacity through three components: a frozen encoder, a slot-based memory router, and a lightweight classifier. Then, they design two mechanisms, i.e., Statistically-Grounded Slot Expansion and Dual-Stage Contrastive–Distillation Adaptation, to enable continual learning on large pre-trained models. This paper was reviewed by four expert reviewers and received ratings: two rejections and two acceptances. The reviewers raised some important concerns, such as dependence on the frozen encoder quality and hyperparameter sensitivity. The authors addressed most of the concerns with additional experimental results and detailed explanations. Even though Reviewers pmFB and zoZH didn't further respond to the rebuttals, I believe their concerns were addressed, and no further outstanding issues remained. Therefore, I recommend accepting this paper.

**Reviewer Concerns:**

This paper was reviewed by four expert reviewers and received ratings: two rejections and two acceptances. The reviewers raised some important concerns, such as dependence on the frozen encoder quality and hyperparameter sensitivity. The authors addressed most of the concerns with additional experimental results and detailed explanations.

**Reviewer Scores:**

This paper was reviewed by four expert reviewers and received ratings: two rejections and two acceptances. Two positive reviewers confirmed that their concerns were addressed. Even though Reviewers pmFB and zoZH didn't further respond to the rebuttals, I believe their concerns were addressed, and no further outstanding issues remained.

---

### Decision · Program_Chairs · 2026-01-26

Accept (Poster)